# LayoutPrompter: Awaken the Design Ability of Large Language Models

**Jiawei Lin**[*]
Xi'an Jiaotong University
kylelin@stu.xjtu.edu.cn

**Jiaqi Guo**
Microsoft
jiaqiguo@microsoft.com

**Shizhao Sun**
Microsoft
shizsu@microsoft.com

**Zijiang James Yang**
Xi'an Jiaotong University
zijiang@xjtu.edu.cn

**Jian-Guang Lou**
Microsoft
jlou@microsoft.com

**Dongmei Zhang**
Microsoft
dongmeiz@microsoft.com

## Abstract

Conditional graphic layout generation, which automatically maps user constraints to high-quality layouts, has attracted widespread attention today. Although recent works have achieved promising performance, the lack of *versatility* and *data efficiency* hinders their practical applications. In this work, we propose Layout-Prompter, which leverages large language models (LLMs) to address the above problems through in-context learning. LayoutPrompter is made up of three key components, namely input-output serialization, dynamic exemplar selection and layout ranking. Specifically, the input-output serialization component meticulously designs the input and output formats for each layout generation task. Dynamic exemplar selection is responsible for selecting the most helpful prompting exemplars for a given input. And a layout ranker is used to pick the highest quality layout from multiple outputs of LLMs. We conduct experiments on all existing layout generation tasks using four public datasets. Despite the simplicity of our approach, experimental results show that LayoutPrompter can compete with or even outperform state-of-the-art approaches on these tasks without any model training or fine-tuning. This demonstrates the effectiveness of this versatile and training-free approach. In addition, the ablation studies show that LayoutPrompter is significantly superior to the training-based baseline in a low-data regime, further indicating the data efficiency of LayoutPrompter. Our project is available here.

## 1   Introduction

*Layout*, which consists of a set of well-arranged graphic elements, plays a critical role in graphic design. To alleviate the workload of designers and allow non-expert users to engage in the design process, numerous studies have delved into the automatic layout generation for diverse user needs [7, 15, 18, 19, 21, 22, 39] (i.e., layout constraints). Based on input layout constraints, existing conditional layout generation tasks can be categorized into the following groups: *constraint-explicit layout generation* (e.g., generating layouts conditioned on element types), *content-aware layout generation*, and *text-to-layout* (see the left side of Figure 1 for constraint examples). Early works in this field [7, 19, 21, 22] primarily focus on individual tasks and develop task-specific model architectures and optimization methods. More recently, task-generic approaches [15, 12, 14] have emerged. Compared to task-specific methods, they achieve greater flexibility and controllability on more tasks, while maintaining the quality of the generated layouts.

---

[*]Work done during an internship at Microsoft Research Asia.

37th Conference on Neural Information Processing Systems (NeurIPS 2023).

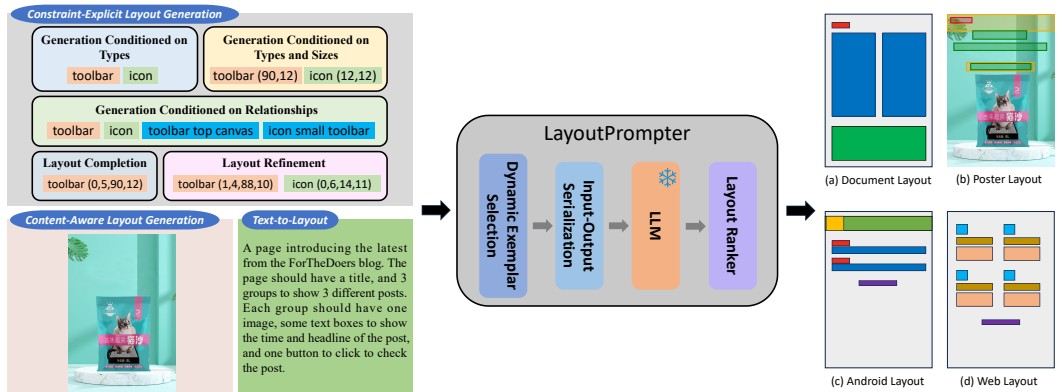

Figure 1: LayoutPrompter is a versatile method for graphic layout generation, capable of solving various conditional layout generation tasks (as illustrated on the left side) across a range of layout domains (as illustrated on the right side) without any model training or fine-tuning.

Although state-of-the-art methods [15, 12, 14, 9, 24] have achieved promising results, they still suffer from some limitations that impede their applications in real-world scenarios. First, *the previous approaches struggle to simultaneously cope with all the layout generation tasks depicted in Figure 1*. They are typically tailored for specific tasks and cannot be applied to others. For instance, the state-of-the-art diffusion-based model LayoutDM [14] proposes to inject *explicit* layout constraints through masking or logit adjustment during inference, but it fails to do so for implicit or vague constraints, e.g., constraints expressed in natural language (i.e., text-to-layout). Consequently, distinct models need to be deployed for different tasks, leading to inconvenience. This motivates us to explore a more versatile approach for layout generation. Second, *the existing methods are not data-efficient either*. They usually necessitate extensive constraint-layout pair data for model training. For example, LayoutFormer++ [15] relies on the publaynet dataset [40] with a size of 300K for training to generate aesthetically pleasing document layouts. However, collecting such large datasets for some low-resource layout domains (e.g., poster layouts) is prohibitively expensive. Besides, even if such a dataset is available, training is time-consuming and costly. Hence, there is a pressing need to develop a data-efficient layout generation method.

In this work, we consider leveraging the powerful pre-trained large language models (LLMs) to address the above problems. The intuition behind is as follows. First, recent research has shown the versatility of LLMs in various tasks [28, 13, 1, 38]. By carefully designing input-output formats, these tasks can be converted into sequence-to-sequence generation problems and effectively addressed by LLMs. This emerging trend inspires us to utilize LLMs to tackle all conditional layout generation tasks in a unified manner. Second, since the training corpus contains layout source code [28, 4] (e.g., HTML code and XML code), LLMs have acquired some layout-related knowledge during pre-training. For example, they may inherently possess the ability to align graphic elements and avoid unnecessary overlap between them, which is beneficial for producing high-quality and visually appealing layouts. Consequently, an LLMs-based approach holds promise to enhance data efficiency compared to existing models that are trained from scratch. Third, an additional advantage of LLMs lies in their remarkable in-context learning performance [3, 28, 36, 35, 13]. It means that instead of fine-tuning LLMs individually for each layout generation task, we can simply prompt them to perform the desired task with a few input-output demonstrations. This characteristic further allows LLMs to generate layouts in a training-free manner without any parameter updates.

To this end, we propose *LayoutPrompter* (see Figure 1). It formulates all conditional layout generation tasks as sequence-to-sequence transformation problems and leverages LLMs to tackle them through in-context learning. To unleash the full potential of LLMs for layout generation, two key issues need to be addressed. First, how to awaken the layout-related knowledge in LLMs for achieving decent performance? Second, how to facilitate LLMs understanding diverse user constraints and layout characteristics in distinct domains? LayoutPrompter tackles the two issues with the *input-output serialization* module and the *dynamic exemplar selection* module, respectively. **I. Input-Output Serialization.** Since prevalent LLMs can only read token sequences, this module is responsible for representing user constraints and layouts as sequences so that LLMs can sufficiently exploit their

| Methods | Versatile | Data-Efficient | Training-Free |
|---|---|---|---|
| LayoutTransformer [7], BLT [19], and so on [9, 24, 16, 30] | ✘ | ✘ | ✘ |
| LayoutFormer++ [15], LayoutDM [14], LGDM [12] | *partially* | ✘ | ✘ |
| LayoutPrompter (ours) | ✔ | ✔ | ✔ |

Table 1: A comparison between existing conditional layout generation methods and LayoutPrompter.

related knowledge. To represent input layout constraints as sequences, we borrow the successful experience of LayoutFormer++ [15], where they present two simple but effective principles (i.e., constraint representation and constraint combination) to serialize constraints. We experimentally find that the serialization scheme is also effective for LLMs. To represent layouts as sequences, our principle is to convert them into a format resembling what LLMs have encountered during pre-training, thereby leveraging the existing layout-related knowledge within LLMs. Specifically, we serialize the layout into the corresponding source code (e.g., HTML) to obtain the output sequence. **II. Dynamic Exemplar Selection.** This module is used to select prompting exemplars that have similar layout constraints to the test samples. In contrast to random exemplars, dynamic exemplars ensure that LLMs receive the most relevant context, so they can better comprehend the desired constraints and produce plausible layouts accordingly. To support this technique, we develop an evaluation suite to measure the constraint similarities between a given test sample and all candidate exemplars from the training set. Then, we select those with the highest similarity scores as prompting exemplars. In addition, we introduce a layout ranker to further improve LayoutPrompter's performance. Considering that LLMs can produce distinct outputs through sampling, we generate multiple layouts with the same input constraints, and use the ranker to select the highest-quality one as the final output.

We conduct extensive experiments on various tasks and layout domains to evaluate LayoutPrompter. Experimental results show that LayoutPrompter can tackle all existing conditional layout generation tasks, demonstrating its versatility. Despite without any model training or fine-tuning, LayoutPrompter is on par or even better than the state-of-the-art approaches. Besides, our ablation studies exhibit that LayoutPrompter can still achieve good performance when there is only a small set of candidate exemplars, indicating that it is superior to existing training-based methods in terms of data efficiency. In summary, LayoutPrompter is a versatile, data-efficient and training-free layout generation method.

## 2 Related Work

**Graphic Layout Generation.** Automatic graphic layout generation is an emerging research topic in recent years. To meet diverse user requirements, existing methods have defined various layout generation tasks, including layout generation conditioned on element types [21, 19, 18], layout generation conditioned on element types and sizes [19], layout generation conditioned on element relationships [18, 21], layout completion [7, 23] and refinement [30]. In addition to these constraint-explicit tasks, some works consider more challenging but useful tasks, such as content-aware layout generation [39, 9] and text-to-layout [11, 24]. Content-aware layout generation aims at arranging spatial space for pre-defined elements on a given canvas. The generated layouts not only need to be visually pleasing, but also avoid salient areas of the canvas. Text-to-layout is to generate layouts according to human language descriptions.

Early works in this field primarily focus on an individual task and propose task-specific approaches based on Generative Adversarial Networks (GANs) [22, 9], Variational Autoencoders (VAEs) [21, 16] and Transformers [19, 7, 11, 24, 30]. Recently, some general approaches [15, 12, 14, 37] have appeared. LayoutFormer++ [15] proposes to represent various constraints as sequences and then leverages a Transformer [32] encoder-decoder architecture to generate layouts from constraint sequences. [12, 14] develop diffusion-based models for constraint-explicit layout generation. However, none of the existing methods can simultaneously handle all layout generation tasks. Furthermore, these methods are highly dependent on large amounts of training data, which hinders their practical applications. In this work, we introduce techniques such as dynamic exemplar selection, input-output serialization and layout ranking to effectively utilize LLMs to overcome the above limitations, making LayoutPrompter a versatile and data-efficient approach (see Table 1).

**Large Language Models.** Large language models (LLMs) with billions of parameters, such as GPT [3, 28], PaLM [6] and LLaMa [31], have demonstrated excellent few-shot performance on

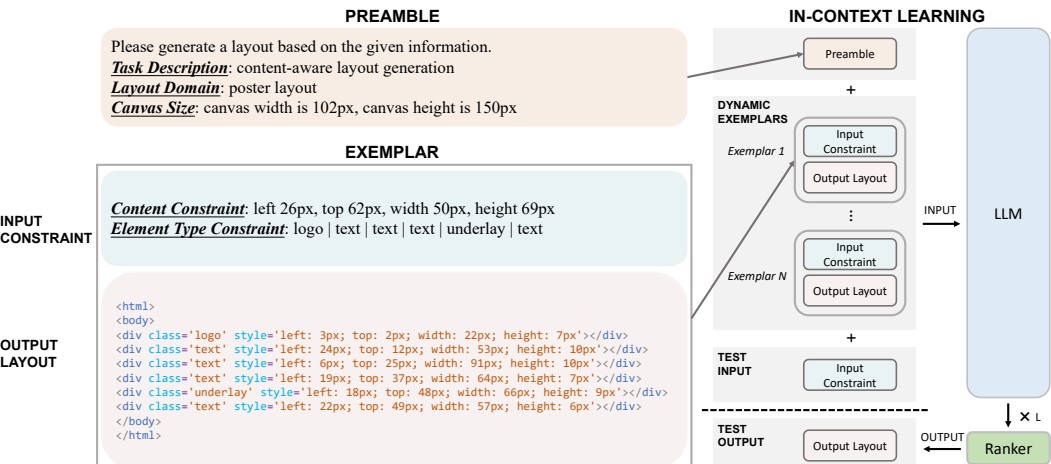

Figure 2: An overview of LayoutPrompter. The complete prompt consists of a task-specific preamble, $N$ in-context exemplars and a test input. The exemplars are dynamically retrieved from the training set according to the test input. Subsequently, the prompt is fed into an LLM to generate $L$ distinct layouts. We employ a layout ranker to select the best one as the final output.

various natural language processing (NLP) tasks. Thanks to the emergent ability [33] brought by the scale of model and data, they largely outperform prior supervised approaches and even match human-level performance on some tasks, without any finetuning. The versatility and effectiveness of LLMs inspire us to develop a layout generation method based on them.

Recent studies show that the prompting strategy plays a crucial role in model performance. For example, chain-of-thought (CoT) prompting [34] is proposed to improve the reasoning ability of LLMs by incorporating intermediate reasoning steps in the exemplars. Least-to-most prompting [41, 17] (also known as decomposed prompting) is introduced to solve complex multi-step reasoning tasks. To enhance contextual knowledge, [26, 10] use a retrieval module to dynamically select in-context exemplars. They experimentally find that exemplars semantically similar to test samples can better unleash the model's knowledge. Specifically, they use a sentence encoder to convert model inputs to vector representations. Then, for each test sample, they retrieve the nearest neighbors in the encoded sentence embedding space to construct prompts. Motivated by them, we propose a similar prompting strategy in this work. Since the input of layout generation tasks is different from prior works, we introduce a customized evaluation suite to measure sample distances. Experimental results demonstrate its effectiveness in LayoutPrompter.

## 3 LayoutPrompter

In this section, we elaborate on LayoutPrompter, a versatile, data-efficient and training-free layout generation method built upon LLMs. Our main contribution lies in proposing a set of useful techniques for applying LLMs to layout generation. Specifically, to support sequence-to-sequence transformation and make maximum use of the design knowledge within LLMs, we carefully consider the serialization scheme that represents task inputs and outputs as sequences (Section 3.2). Moreover, to enhance the comprehension of user-specified layout constraints, we propose a dynamic exemplar selection module to retrieve the most relevant exemplars from the training set to perform in-context learning (Section 3.3). Besides, a layout ranker is designed to evaluate layout quality and rank multiple layouts generated under the same constraints, further improving model performance (Section 3.4).

### 3.1 Overview

Let's consider a conditional layout generation task. We denote its training set as $\mathcal{D} = \{(x_j, y_j)\}_{j=1}^{M}$. Here, $(x_j, y_j)$ represents the $j$-th sample of $\mathcal{D}$, which is an (input constraint, output layout) pair, and $M$ is the total number of samples. As illustrated in Figure 2, for a given test query $x_{\text{test}}$, the in-context learning prompt $\mathbf{P}$ is composed by sequentially concatenating a task-specific preamble $R$,

$N$ exemplars and the query itself:

$$\mathbf{P} = [R; F_X(x_{k_1}); F_Y(y_{k_1}); \ldots; F_X(x_{k_N}); F_Y(y_{k_N}); F_X(x_{\text{test}})], \quad \{k_i\}_{i=1}^N = G(x_{\text{test}}, \mathcal{D}). \quad (1)$$

To be more specific, the preamble $R$ provides the essential information about the target task, such as the *task description*, *layout domain* and *canvas size*. $F_X(\cdot)$ and $F_Y(\cdot)$ are serialization functions that transform task input $x$ and output $y$ into sequences, respectively. $G(\cdot, \cdot)$ denotes an exemplar selection function, which retrieves the in-context exemplars from $\mathcal{D}$ according to $x_{\text{test}}$. The details of $F_X$, $F_Y$ and $G$ will be elaborated in the following sections.

Notably, when the number of exemplars $N$ is set to 0, few-shot in-context learning degenerates to zero-shot learning, where LLMs predict the test output $y_{\text{test}}$ solely based on the preamble $R$ and $x_{\text{test}}$. In our experiments (see Section B.2 in Appendix), we find that additional exemplar guidance can help LLMs better comprehend the task and grasp the rough pattern of the required layouts. Hence, we opt for few-shot learning ($N > 0$) instead of zero-shot learning ($N = 0$) in this work.

## 3.2 Input-Output Serialization

To begin, we first establish some notations. For each element $e$ that constitutes a layout, we describe it by its element type $c$, left coordinate $l$, top coordinate $t$, width $w$ and height $h$, i.e., $e = (c, l, t, w, h)$. Here, $c$ is a categorical attribute. The other four are numerical geometric attributes, which will be discretized in the implementation (see Section A in Appendix).

**Input Constraint Serialization.** For constraint-explicit layout generation, the input constraints are element-wise constraints on $e$. We serialize such constraints in the same way as LayoutFormer++ [15], where they represent each constraint as a sequence and then combine different constraints through concatenation. For example, if $x$ specifies the element types and sizes, $F_X(x)$ takes the form of $F_X(x) = $ "$c_1 w_1 h_1 | c_2 w_2 h_2 | \ldots$". In this

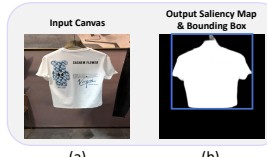

Figure 3: An input canvas is converted into a saliency map.

work, we adopt these ready-made sequences for constraint-explicit layout generation tasks. Regarding content-aware layout generation, the image nature of the input canvas poses a unique challenge for serialization, i.e., enabling LLMs that can only read text to perceive image content. Inspired by DS-GAN [9], we recognize that the saliency map [8] can well capture the key content shape of a canvas while discarding other high-frequency, irrelevant details (see Figure 3). To facilitate serialization, we further convert it into a rectified saliency map $m = (l_m, t_m, w_m, h_m)$ by detecting region boundaries with pixel values greater than a certain threshold. After preprocessing, the input canvas $x$ can be represented in a format understandable by LLMs: $F_X(x) = $ "Content Constraint: left $l_m$px,top $t_m$px,width $w_m$px,height $h_m$px". For the text-to-layout task, where natural language descriptions are used to generate layouts, the constraint sequence is simply the input text itself.

**Output Layout Serialization.** For the output $y$, we propose to serialize it into the HTML format that LLMs are more familiar with and good at, rather than the plain sequence used in prior works [15, 7]. Following common HTML representation, we denote the complete output sequence as a concatenation of multiple HTML segments $a$: $F_Y(y) = [a_1; a_2; \ldots]$. Here, the $i$-th segment $a_i$ represents the $i$-th graphic element $e_i$ of $y$. It specifies the element attributes in the following format:

<div class="$c_i$" style="left:$l_i$px; top:$t_i$px; width:$w_i$px; height:$h_i$px"></div>. (2)

Thanks to the powerful in-context learning ability of LLMs, the test output $y_{\text{test}}$ will be predicted in the same HTML format, making it easy to extract the required element attributes from the output. More input-output examples can be found in Section D of the supplementary material.

## 3.3 Dynamic Exemplar Selection

As mentioned above, $G$ selects $N$ in-context exemplars that have the most similar layout constraints to $x_{\text{test}}$ from $\mathcal{D}$. The selected exemplars are randomly shuffled and combined to construct $\mathbf{P}$ (see Equation 1), thereby enhancing LLMs' understanding of various constraints. To achieve this, we design an evaluation suite $s$ to measure the constraint similarity between the test query $x_{\text{test}}$ and each

| Tasks | Methods | RICO | | | | | PubLayNet | | | | |
|---|---|---|---|---|---|---|---|---|---|---|---|
| | | mIoU ↑ | FID ↓ | Align. ↓ | Overlap ↓ | Vio. % ↓ | mIoU ↑ | FID ↓ | Align. ↓ | Overlap ↓ | Vio. % ↓ |
| Gen-T | BLT | 0.216 | 25.633 | 0.150 | 0.983 | - | 0.140 | 38.684 | 0.036 | 0.196 | - |
| | LayoutFormer++ | 0.432 | 1.096 | 0.230 | 0.530 | 0. | 0.348 | 8.411 | 0.020 | 0.008 | 0. |
| | LayoutPrompter | 0.429 | 3.233 | **0.109** | **0.505** | 0.64 | **0.382** | **3.022** | 0.037 | 0.047 | 0.50 |
| Gen-TS | BLT | 0.604 | 0.951 | 0.181 | 0.660 | 0. | 0.428 | 7.914 | 0.021 | 0.419 | 0. |
| | LayoutFormer++ | 0.620 | 0.757 | 0.202 | 0.542 | 0. | 0.471 | 0.720 | 0.024 | 0.037 | 0. |
| | LayoutPrompter | 0.552 | 1.458 | **0.145** | 0.544 | 0.18 | 0.453 | 1.067 | 0.049 | 0.091 | 0. |
| Gen-R | CLG-LO | 0.286 | 8.898 | 0.311 | 0.615 | 3.66 | 0.277 | 19.738 | 0.123 | 0.200 | 6.66 |
| | LayoutFormer++ | 0.424 | 5.972 | 0.332 | 0.537 | 11.84 | 0.353 | 4.954 | 0.025 | 0.076 | 3.9 |
| | LayoutPrompter | 0.400 | **5.178** | **0.101** | 0.564 | 10.58 | 0.347 | **3.620** | 0.037 | 0.161 | 12.29 |
| Completion | LayoutTransformer | 0.363 | 6.679 | 0.194 | 0.478 | - | 0.077 | 14.769 | 0.019 | 0.0013 | - |
| | LayoutFormer++ | 0.732 | 4.574 | 0.077 | 0.487 | - | 0.471 | 10.251 | 0.020 | 0.0022 | - |
| | LayoutPrompter | 0.667 | 7.318 | 0.084 | **0.428** | - | 0.476 | **2.132** | 0.023 | 0.017 | - |
| Refinement | RUITE | 0.811 | 0.107 | 0.133 | 0.483 | - | 0.781 | 0.061 | 0.029 | 0.020 | - |
| | LayoutFormer++ | 0.816 | 0.032 | 0.123 | 0.489 | - | 0.785 | 0.086 | 0.024 | 0.006 | - |
| | LayoutPrompter | 0.745 | 0.978 | 0.159 | **0.478** | - | 0.647 | 0.278 | 0.072 | 0.048 | - |

Table 2: Quantitative comparison with baselines on constraint-explicit layout generation tasks. ↑ indicates larger values are better, ↓ indicates smaller values are better.

candidate exemplar $(x_j, y_j) \in \mathcal{D}$. Then, $G$ can be further expressed as a Top-k selection function:

$$G(x_{\text{test}}, \mathcal{D}) \triangleq \text{Top-k}(\bigcup_{(x_j, y_j) \in \mathcal{D}} \{s(x_{\text{test}}, x_j)\}, N). \qquad (3)$$

Since we divide existing layout generation tasks into three categories, each with distinct input constraints, their similarity measures have different representations. We'll elaborate below.

**Constraint-Explicit Layout Generation.** As constraint-explicit layout generation tasks only consider element-wise constraints, we define $s(x_{\text{test}}, x_j)$ using inter-element constraint similarities. Specifically, we construct a bipartite graph between $x_{\text{test}} = \{p_{\text{test}}^u\}_{u=1}^U$ and $x_j = \{p_j^v\}_{v=1}^V$, where $p$ denotes the element-wise constraint on $e$. $U, V$ are the constraint numbers of $x_{\text{test}}, x_j$. Then, the inter-element similarity $W$ (i.e., the weight of bipartite graph) and the overall constraint similarity $s$ are defined as:

$$s(x_{\text{test}}, x_j) \triangleq \frac{1}{|\mathbb{M}_{\max}|} \sum_{(p_{\text{test}}^u, p_j^v) \in \mathbb{M}_{\max}} W(p_{\text{test}}^u, p_j^v), \quad W(p_{\text{test}}^u, p_j^v) = \mathbb{1}(p_{\text{test}}^u, p_j^v) 2^{-\|\mathbf{g}_{\text{test}}^u - \mathbf{g}_j^v\|_2}. \qquad (4)$$

Here, $\mathbb{1}$ is a 0-1 function equal to 1 if $p_{\text{test}}^u$ and $p_j^v$ specify the same element type, and 0 otherwise. This ensures that constraint similarity is only considered between elements with the same type. $\mathbf{g}_{\text{test}}^u$ and $\mathbf{g}_j^v$ are specified geometric attributes of $p_{\text{test}}^u$ and $p_j^v$. Given the edge weight $W$ of the bipartite graph, we adopt Hungarian method [20] to obtain the maximum matching $\mathbb{M}_{\max}$. And $s(x_{\text{test}}, x_j)$ is calculated as the average weight of matched edges (as shown in Equation 4).

**Content-Aware Layout Generation.** The constraint of content-aware layout generation is the input canvas. The similarity of two canvases $x_{\text{test}}, x_j$ is defined as the IoU (Intersection over Union) of their rectified saliency maps (see Section 3.2) $m_{\text{test}}, m_j$:

$$s(x_{\text{test}}, x_j) \triangleq \text{IoU}(m_{\text{test}}, m_j) = \frac{|m_{\text{test}} \cap m_j|}{|m_{\text{test}} \cup m_j|}. \qquad (5)$$

**Text-to-Layout.** We leverage the CLIP [29] text encoder to encode input texts into embeddings. The constraint similarity $s(x_{\text{test}}, x_j)$ is defined as the cosine similarity of input text embeddings $n_{\text{test}}, n_j$:

$$s(x_{\text{test}}, x_j) \triangleq \frac{n_{\text{test}} \cdot n_j}{\|n_{\text{test}}\| \|n_j\|}. \qquad (6)$$

### 3.4 Layout Ranker

People usually judge the quality of generated layouts from two perspectives: (1) whether they are visually pleasing; (2) whether they look like the real layouts. Therefore, our proposed layout ranker follows the same principles to evaluate layout quality. To be more specific, it measures the quality of an output layout using a combination of metrics:

$$q(y_{\text{test}}) = \lambda_1 \text{Alignment}(y_{\text{test}}) + \lambda_2 \text{Overlap}(y_{\text{test}}) + \lambda_3 (1 - \text{mIoU}(y_{\text{test}})). \qquad (7)$$

| Dataset | Domain | Associated Task | # Training Set | # Test Set | # Element Types |
|---|---|---|---|---|---|
| RICO | Android | constraint-explicit layout generation | 31,694 | 3,729 | 25 |
| PubLayNet | document | constraint-explicit layout generation | 311,397 | 10,998 | 5 |
| PosterLayout | poster | content-aware layout generation | 9,974 | 905 | 3 |
| WebUI | web | text-to-layout | 3,835 | 487 | 10 |

Table 3: Dataset statistics. Note that these datasets are only used on specific tasks.

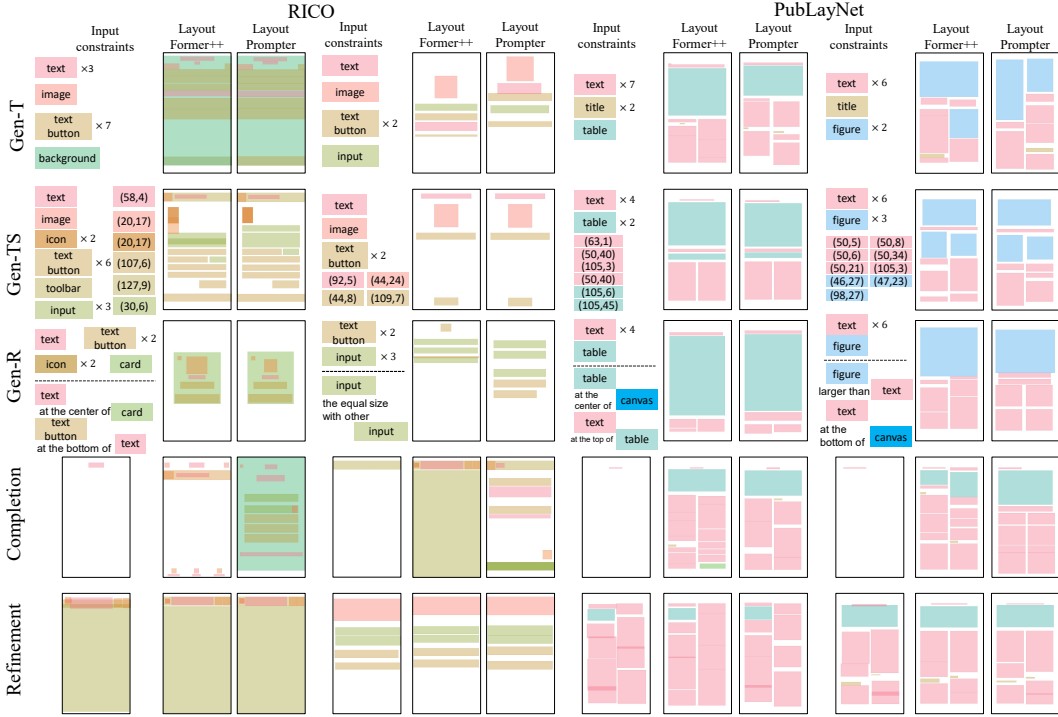

Figure 4: Qualitative comparison between LayoutPrompter and the state-of-the-art baseline Layout-Former++ [15] on constraint-explicit layout generation tasks (better view in color and 2× zoom).

Here, $\lambda_1$, $\lambda_2$ and $\lambda_3$ are hyper-parameters to balance the importance of each metric. Alignment and Overlap reflect quality from the perspective (1), while mIoU mainly focuses on perspective (2). We will introduce them in Section 4.1. The output layout with the lowest $q$ value (lower $q$ indicates better quality) is returned as the final output.

## 4 Experiments

### 4.1 Setups

**Datasets.** We conduct experiments on 4 datasets, including RICO [27], PubLayNet [40], Poster-Layout [9] and WebUI [24]. Their statistics and usages are illustrated in Table 3. For RICO and PubLayNet, we adopt the same dataset splits as LayoutFormer++ [15]. While for PosterLayout, the training set includes 9,974 poster-layout pairs, and the remaining 905 posters are used for testing. Regarding the WebUI dataset, we adopt the dataset splits provided by parse-then-place [24]. In all cases, the in-context exemplars are retrieved from the full training set.

**Baselines.** Since constraint-explicit layout generation tasks have task-specific and task-generic methods, we compare LayoutPrompter against both kinds of state-of-the-art methods on these tasks. Concretely, we choose LayoutFormer++ [15] as the common task-generic baseline. The task-specific baselines are (1) BLT [19] for generation conditioned on types (Gen-T), (2) BLT [19] for generation conditioned on types with sizes (Gen-TS), (3) CLG-LO [18] for generation conditioned on relationships (Gen-R), (4) LayoutTransformer [7] for completion, and (5) RUITE [30] for refinement. Moreover, we compare LayoutPrompter with DS-GAN [9] and CGL-GAN [42] on content-aware layout generation. We compare with Mockup [11] and parse-then-place [24] on text-to-layout.

| | Val ↑ | Ove ↓ | Ali ↓ | Und$_l$ ↑ | Und$_s$ ↑ | Uti ↑ | Occ ↓ | Rea ↓ |
|---|---|---|---|---|---|---|---|---|
| CGL-GAN | 0.7066 | 0.0605 | 0.0062 | 0.8624 | 0.4043 | 0.2257 | 0.1546 | 0.1715 |
| DS-GAN | 0.8788 | 0.0220 | 0.0046 | 0.8315 | 0.4320 | 0.2541 | 0.2088 | 0.1874 |
| LayoutPrompter (Ours) | **0.9992** | **0.0036** | **0.0036** | **0.8986** | **0.8802** | **0.2597** | **0.0992** | 0.1723 |

Table 4: Quantitative comparison with baselines on content-aware layout generation task.

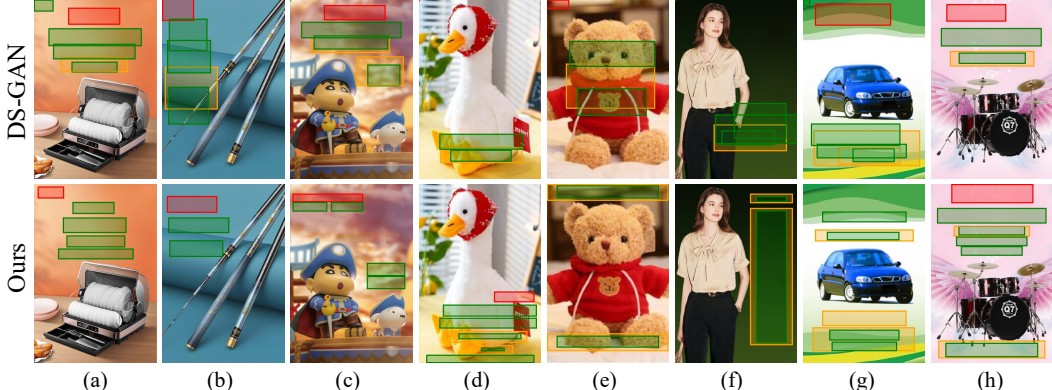

(a)  (b)  (c)  (d)  (e)  (f)  (g)  (h)

Figure 5: Qualitative results generated by DS-GAN and LayoutPrompter on content-aware layout generation. There are three element types, including logo (red), text (green) and underlay (yellow).

**Evaluation Metrics.** To evaluate the performance of LayoutPrompter and baselines, we use the following quantitative metrics. For constraint-explicit layout generation and text-to-layout, we employ four standard metrics. *Alignment (Align.)* [22] gauges how well the elements in a layout are aligned with each other. *Overlap* [22] computes the overlapping area between two arbitrary elements in a layout. *Maximum IoU (mIoU)* [18] calculates the highest Intersection over Union (IoU) between a generated layout and real layouts. *Fréchet Inception Distance (FID)* [18] measures how similar the distribution of the generated layouts is to that of real layouts. Additionally, we introduce another metric *Constraint Violation Rate (Vio. %)* [15] to evaluate how well the generated layouts satisfy their input constraints. It is the ratio of violated constraints to all constraints. In text-to-layout, as a textual description may involve the type, position and size constraints of elements, we follow parse-then-place [24] and further break down this metric into *Type Vio. %* and *Pos & Size Vio. %*. As for content-aware layout generation, we adopt the eight metrics defined in DS-GAN [9]. Some of them belong to graphic metrics, such as *Val*, *Ove*, *Ali*, *Und$_l$* and *Und$_s$*. Others are content-aware metrics, including *Uti*, *Occ* and *Rea*. Please refer to [9] for more details.

**Implementation Details.** In this work, we conduct experiments on GPT-3 [3] `text-davinci-003` model. We place $N = 10$ exemplars in the prompt $\mathbf{P}$. For each test sample, we generate $L = 10$ different outputs $y_{\text{test}}$. The hyper-parameters involved in the layout ranker module are set to $\lambda_1 = 0.2, \lambda_2 = 0.2$, and $\lambda_3 = 0.6$. When running GPT-3, we fix the parameters to the default values of the OpenAI API, where the sampling temperature is 0.7 and the penalty-related parameters are set to 0.

### 4.2 Main Results

Tables 2, 4, 5 and Figures 4, 5, 6 show the quantitative and qualitative results on various layout generation tasks (see more qualitative results in Section C of the supplementary material). Although LayoutPrompter has not undergone model training and fine-tuning, the experimental results demonstrate that it can achieve comparable or even better performance than baselines, which proves that LayoutPrompter is a versatile and training-free layout generation approach. Below, we conduct a detailed analysis of the experimental results.

**Constraint-Explicit Layout Generation.** Table 2 shows the quantitative results. On each constraint-explicit layout generation task, LayoutPrompter is compared with a task-specific method and another common baseline, LayoutFormer++ [15]. Although not trained on these downstream tasks, LayoutPrompter still exhibits competitive quantitative results. Furthermore, it even outperforms the baselines on some metrics (e.g., Align. and Overlap on Gen-T task, RICO dataset). The corresponding qualitative results are shown in Figure 4. Here, we only compare with the state-of-the-art baseline (measured by quantitative metrics), LayoutFormer++. The qualitative comparison indicates

| | mIoU ↑ | FID ↓ | Align. ↓ | Overlap ↓ | Type Vio. % ↓ | Pos & Size Vio. % ↓ |
|---|---|---|---|---|---|---|
| Mockup | 0.1927 | 37.0123 | 0.0059 | 0.4348 | 31.49 | 44.92 |
| parse-then-place | 0.6841 | 2.9592 | 0.0008 | 0.1380 | 11.36 | 19.14 |
| LayoutPrompter (Ours) | 0.3190 | 10.7706 | 0.0009 | **0.0892** | 15.09 | 23.78 |

Table 5: Quantitative comparison with baselines on text-to-layout.

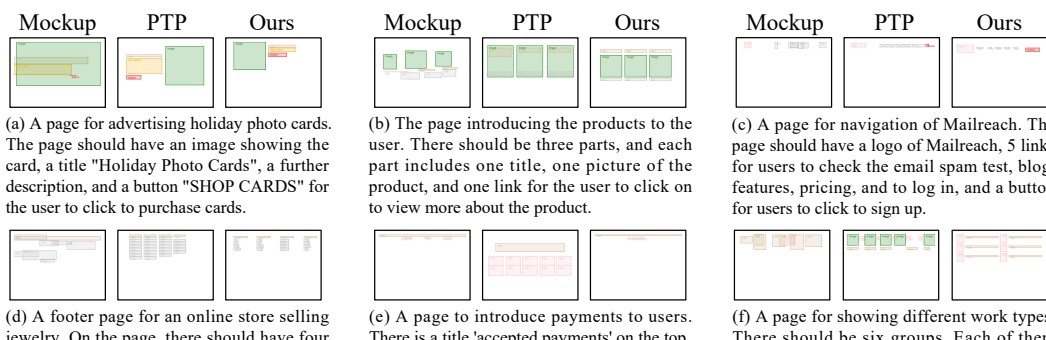

(a) A page for advertising holiday photo cards. The page should have an image showing the card, a title "Holiday Photo Cards", a further description, and a button "SHOP CARDS" for the user to click to purchase cards.

(b) The page introducing the products to the user. There should be three parts, and each part includes one title, one picture of the product, and one link for the user to click on to view more about the product.

(c) A page for navigation of Mailreach. The page should have a logo of Mailreach, 5 links for users to check the email spam test, blog, features, pricing, and to log in, and a button for users to click to sign up.

(d) A footer page for an online store selling jewelry. On the page, there should have four lists of links for more info. In each list, there should be a title and 4-6 links.

(e) A page to introduce payments to users. There is a title 'accepted payments' on the top. There are also four logos of payment companies under it.

(f) A page for showing different work types. There should be six groups. Each of them includes one logo and one text which shows the title of the work.

Figure 6: Qualitative results of Mockup, parse-then-place (short as PTP) and LayoutPrompter on text-to-layout (better view in color and 2× zoom).

that LayoutPrompter achieves as good controllability and generation quality as LayoutFormer++. First, the layouts generated by our approach satisfy various input constraints well, including type constraints, size constraints, relationship constraints, etc. Second, our approach can also produce visually pleasing layouts with well-aligned elements and small overlapping areas. Both qualitative and quantitative results demonstrate the effectiveness of LayoutPrompter.

**Content-Aware Layout Generation.** The quantitative and qualitative results are presented in Table 4 and Figure 5, respectively. Remarkably, LayoutPrompter surpasses the training-based baselines on almost all metrics. This indicates that LayoutPrompter is capable of producing higher-quality and more content-aware layouts compared to the baselines. The rendered results further validate the conclusion. For example, in columns (f) and (g) of Figure 5, the layouts from DS-GAN [9] contain serious misalignments and overlaps. And column (e) shows that DS-GAN sometimes fails to generate content-aware layouts. In contrast, our approach can not only produce aesthetic layouts but also avoid the most salient objects in the input canvas, such as the person, teddy bear, car, etc.

**Text-to-Layout.** The quantitative and qualitative comparisons are shown in Table 5 and Figure 6. Since text-to-layout is one of the most challenging layout generation tasks, LayoutPrompter slightly lags behind the current state-of-the-art method parse-then-place [24], especially on mIoU and FID metrics. However, on the other four metrics, LayoutPrompter is comparable to the baselines. Thanks to the excellent understanding capability of LLMs, our approach can better satisfy the constraints specified in textual descriptions in some cases. For example, in cases (d) and (e) of Figure 6, LayoutPrompter successfully generates *4-6 links* and *four logos*, while parse-then-place makes wrong predictions about the number of elements.

## 4.3 Ablation Studies

**Effect of Introduced Components.** LayoutPrompter has three key components, including input-output serialization, dynamic exemplar selection and layout ranking. To investigate their effects, we perform the following ablation studies (see Table 6). (1) Since LayoutFormer++ [15] has proven the effectiveness of constraint sequences relative to other formats, we only study the effect of HTML representation, which is not covered in previous works. Specifically, we replace HTML with a plain sequence proposed by LayoutFormer++ [15] (denoted as w/o HTML) to represent the output layout. This results in a significant drop in FID and overlap metrics on Gen-T. (2) To understand the contribution of dynamic exemplar selection, we compare against its variant (w/o dynamic selection) that adopts random sampling for exemplar retrieval. LayoutPrompter achieves significantly better FID and mIoU across the board. Though the variant has better Align. and Overlap scores in some tasks, its noticeably poor FID and mIoU scores indicate that it fails to acquire the layout patterns in specific domains (e.g., the generated layout does not look like a real UI layout). (3) To understand the

|  |  | RICO | | | | |
|---|---|---|---|---|---|---|
| Tasks | Methods | mIoU ↑ | FID ↓ | Align. ↓ | Overlap ↓ | Vio. % ↓ |
| Gen-T | LayoutPrompter | 0.429 | 3.233 | 0.109 | 0.505 | 0.64 |
|  | *w/o HTML* | 0.460 | 7.009 | 0.106 | 0.663 | 0. |
|  | *w/o dynamic selection* | 0.251 | 8.154 | 0.053 | 0.399 | 0.24 |
|  | *w/o layout ranker* | 0.367 | 3.149 | 0.142 | 0.498 | 0.45 |
| Gen-TS | LayoutPrompter | 0.552 | 1.458 | 0.145 | 0.544 | 0.18 |
|  | *w/o dynamic selection* | 0.337 | 8.107 | 0.199 | 0.400 | 0.24 |
|  | *w/o layout ranker* | 0.505 | 1.528 | 0.153 | 0.549 | 0.13 |
| Gen-R | LayoutPrompter | 0.400 | 5.178 | 0.101 | 0.564 | 10.58 |
|  | *w/o dynamic selection* | 0.223 | 14.177 | 0.067 | 0.597 | 15.95 |
|  | *w/o layout ranker* | 0.341 | 5.282 | 0.137 | 0.545 | 6.54 |
| Completion | LayoutPrompter | 0.667 | 7.318 | 0.084 | 0.428 | - |
|  | *w/o dynamic selection* | 0.449 | 17.409 | 0.062 | 0.422 | - |
|  | *w/o layout ranker* | 0.580 | 11.194 | 0.093 | 0.451 | - |
| Refinement | LayoutPrompter | 0.745 | 0.978 | 0.159 | 0.478 | - |
|  | *w/o dynamic selection* | 0.662 | 1.718 | 0.208 | 0.468 | - |
|  | *w/o layout ranker* | 0.705 | 1.161 | 0.188 | 0.478 | - |

Table 6: Ablation studies of the introduced components on RICO.

|  |  | LayoutFormer++ | | | | | LayoutPrompter | | | | |
|---|---|---|---|---|---|---|---|---|---|---|---|
| Tasks | # Training samples | mIoU ↑ | FID ↓ | Align. ↓ | Overlap ↓ | Vio. % ↓ | mIoU ↑ | FID ↓ | Align. ↓ | Overlap ↓ | Vio. % ↓ |
| Gen-T | 500 | 0.176 | 92.643 | 0.272 | 0.668 | 69.27 | 0.343 | 7.201 | 0.105 | 0.539 | 0.11 |
|  | 2,000 | 0.209 | 48.702 | 0.165 | 0.573 | 62.22 | 0.362 | 6.140 | 0.083 | 0.527 | 0.22 |
|  | 10,000 | 0.368 | 3.370 | 0.132 | 0.572 | 11.02 | 0.389 | 4.658 | 0.097 | 0.527 | 0.11 |
|  | Full Set | 0.432 | 1.096 | 0.230 | 0.530 | 0. | 0.429 | 3.233 | 0.109 | 0.505 | 0.64 |
| Gen-TS | 500 | 0.171 | 79.641 | 0.301 | 0.808 | 74.66 | 0.405 | 4.068 | 0.130 | 0.596 | 0.13 |
|  | 2,000 | 0.249 | 39.673 | 0.209 | 0.655 | 53.07 | 0.424 | 3.460 | 0.143 | 0.604 | 0.06 |
|  | 10,000 | 0.529 | 2.395 | 0.215 | 0.596 | 1.86 | 0.464 | 2.606 | 0.138 | 0.580 | 0.06 |
|  | Full Set | 0.620 | 0.757 | 0.202 | 0.542 | 0. | 0.552 | 1.458 | 0.145 | 0.544 | 0.18 |

Table 7: Ablation studies of training set size on RICO.

effect of the proposed layout ranker, we compare it against a variant (w/o layout ranker) that randomly picks a layout from model outputs. We find that the layout ranker consistently yields improvements on the mIoU and Align. metrics of all tasks.

**Effect of Training Set Size.** We switch training set sizes: 500, 2000, 10000 and full set (see Table 7). In our approach, the training set represents the exemplar retrieval pool. The results show that the performance of LayoutFormer++ drops rapidly as the training data decreases, but our method is much slightly affected. When training samples are limited (e.g., 500 and 2000), our approach significantly outperforms the training-based baseline on all metrics. These observations suggest that LayoutPrompter is a more data-efficient approach, which is effective in low-resource scenarios. Due to space limitations, more experimental results on stability, the effect of the number of examples, and generalization ability can be found in Section B of the supplementary material.

## 5    Conclusion and Limitation

In this work, we concentrate on leveraging Large Language Models (LLMs) for conditional layout generation to address issues present in existing methods. To enhance the performance of our approach, we introduce three crucial components: input-output serialization, dynamic exemplar selection, and layout ranking. We conduct experiments on 7 existing layout generation tasks using 4 public datasets. Both qualitative and quantitative results highlight that LayoutPrompter is a versatile, data-efficient, and training-free method capable of generating high-quality, constraint-compliant layouts. Despite these promising results, there are still some limitations. First, the performance of our approach is influenced by the number of elements in the layouts, with more elements leading to more failure cases. Notably, this is not a problem specific to our approach and has been observed in prior work [2] as well. Second, we have not studied whether LayoutPrompter is equally effective for other LLMs such as PaLM and LLaMa. Third, with the rapid development of large multimodal models such as GPT-4V, PaLI [5] and LLaVA [25], we get a promising chance to extend LayoutPrompter to supporting layout constraints specified in a wide range of modalities. We leave them for future research.

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

## A  Coordinate Discretization

In this work, element coordinates are scaled proportionally into a canvas of size $C_W \times C_H$. We follow the baselines to choose these two parameters. Specifically, in RICO, $C_W = 90px$, $C_H = 160px$. In PubLayNet, $C_W = 120px$, $C_H = 160px$. In PosterLayout, $C_W = 102px$, $C_H = 150px$. In WebUI, $C_W = 120px$, $C_H = 120px$. Then, the coordinates are discretized to the nearest integers.

## B  Additional Experimental Results and Analysis

### B.1  Stability of Generation Performance

The output of LLMs varies with the random seed and hyper-parameters (e.g., the temperature). That is, for the same input constraint, LLMs are able to generate many completely different layouts. Since the hyper-parameters are a trade-off between generation quality and diversity, we fix them to the default values of OpenAI API and study the impact of random seeds on model performance. Specifically, we run inference on the test set 10 times, each using a different random seed. Then, we calculate the mean and variance of each quantitative metric (see Table 8). The small variances indicate the stability of LayoutPrompter's performance under different random seeds.

| Tasks | RICO | | | | | PubLayNet | | | | |
|---|---|---|---|---|---|---|---|---|---|---|
| | mIoU ↑ | FID ↓ | Align. ↓ | Overlap ↓ | Vio. % ↓ | mIoU ↑ | FID ↓ | Align. ↓ | Overlap ↓ | Vio. % ↓ |
| Gen-T | 0.368±0.002 | 3.118±0.045 | 0.130±0.010 | 0.498±0.004 | 0.546±0.148 | 0.343±0.001 | 4.014±0.067 | 0.042±0.007 | 0.047±0.002 | 0.490±0.059 |
| Gen-TS | 0.504±0.001 | 1.489±0.037 | 0.155±0.003 | 0.550±0.005 | 0.134±0.025 | 0.393±0.001 | 2.016±0.024 | 0.050±0.008 | 0.098±0.002 | 0. |

Table 8: Effect of random seeds. In this experiment, we disable the layout ranker to eliminate the impact of the ranking mechanism on model performance.

### B.2  Effect of Exemplar Number

We conduct ablation experiments on the number of prompting exemplars. Figure 7 shows the zero-shot ($N = 0$) qualitative results on Gen-T. It is obvious that LLMs fail to generate reasonable layouts in a zero-shot scheme. Table 9 exhibits the quantitative comparison of the Gen-T task on RICO. The results indicate that the number of prompting exemplars mainly affects mIoU and FID. Specifically, as the number of prompting exemplars increases, mIoU and FID get improved. In summary, the number of exemplars has a positive effect on the performance of LayoutPrompter.

PubLayNet                                    RICO

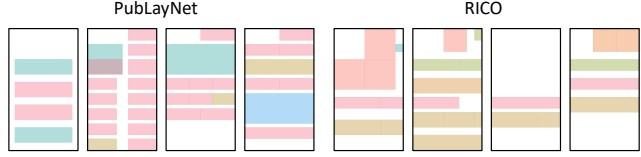

Figure 7: Zero-shot results on the Gen-T task.

| Tasks | # exemplar | RICO | | | | |
|---|---|---|---|---|---|---|
| | | mIoU ↑ | FID ↓ | Align. ↓ | Overlap ↓ | Vio. % ↓ |
| Gen-T | 1 | 0.381 | 5.007 | 0.115 | 0.491 | 0.85 |
| | 3 | 0.413 | 5.098 | 0.120 | 0.492 | 0.51 |
| | 5 | 0.414 | 4.521 | 0.114 | 0.492 | 0.65 |
| | 10 | 0.427 | 3.523 | 0.092 | 0.486 | 0.67 |

Table 9: Ablation studies on the number of prompting exemplars. We run experiments on 1,000 test samples.

### B.3  Generalization Ability

To investigate the generalization ability of LayoutPrompter, we compute the `DocSim` similarity between the generated layouts and their prompting layouts (see Table 10). The `DocSim` of Layout-Former++ is computed between the generated layouts and training layouts. The quantitative results show that LayoutPrompter achieves competitive or even better scores compared to LayoutFormer++,

indicating that LayoutPrompter has a close generalization ability to the training-based method. In addition, we exhibit the qualitative results of the generated layouts and their prompting layouts in Figure 8. The results demonstrate that LayoutPrompter is capable of generating meaningful variations different from the prompting ones.

| Method | RICO | PubLayNet |
|---|---|---|
| LayoutFormer++ | 0.8472 | 0.8266 |
| LayoutPrompter | 0.8563 | 0.8119 |

Table 10: `DocSim` on Gen-T task. The smaller the `DocSim`, the better the generalization ability.

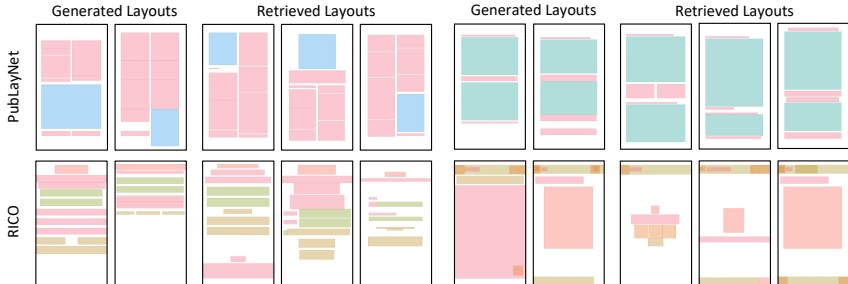

Figure 8: Qualitative results of the generated layouts and corresponding retrieved layouts on Gen-T task. Each case contains 2 generated layouts.

# C    Additional Qualitative Results

In this section, we present additional qualitative results of LayoutPrompter. These results further demonstrate the versatility and effectiveness of our method, which can generate high-quality and constraint-compliant layouts on multiple layout generation tasks.

## C.1    Generation Conditioned on Element Types

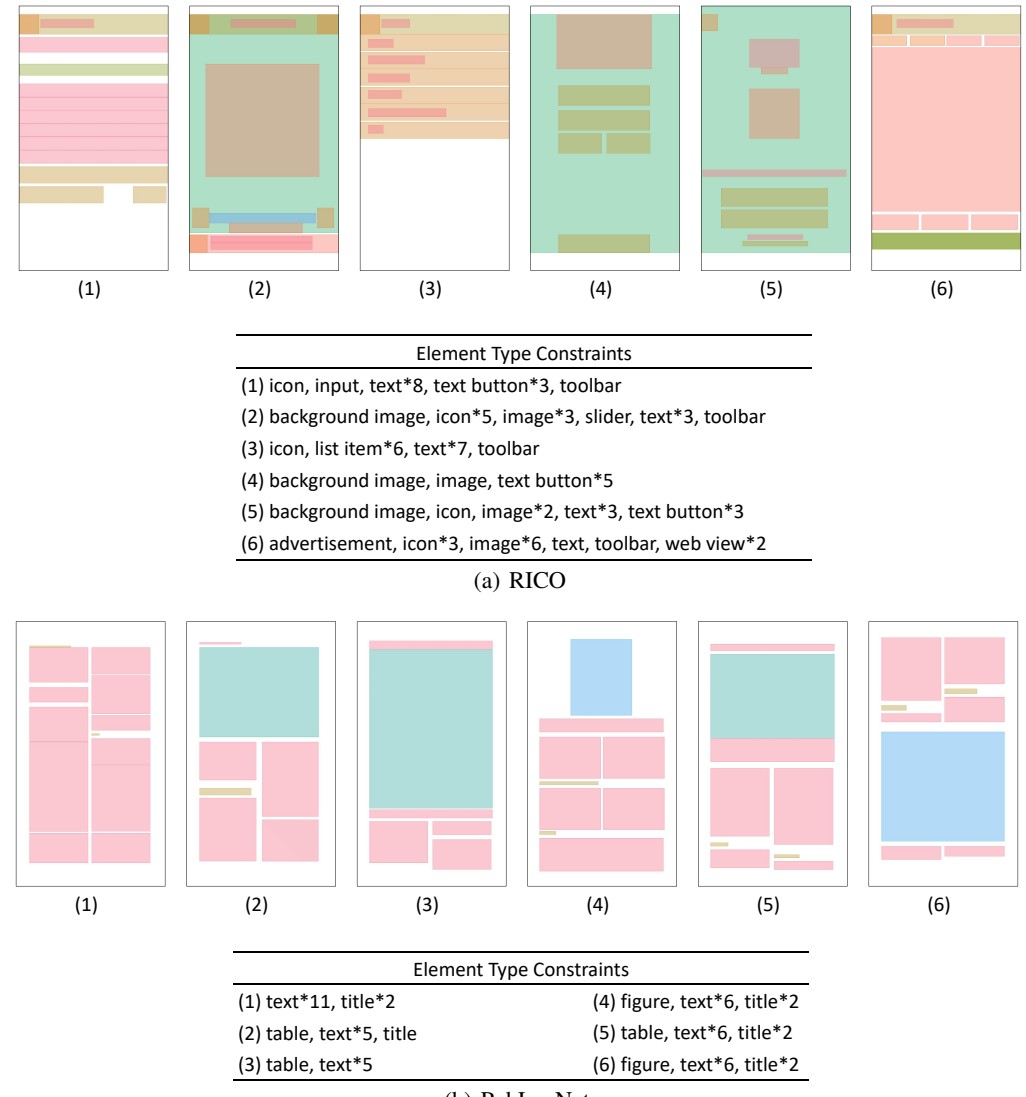

Figure 9: Qualitative results of Gen-T on RICO and PubLayNet. The element type constraints are in the table.

## C.2  Generation Conditioned on Element Types and Sizes

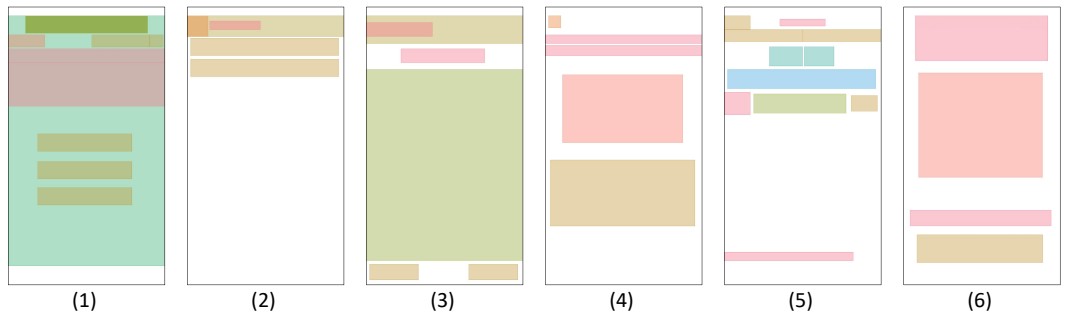

| | | | | | |
|---|---|---|---|---|---|
| (1) | (2) | (3) | (4) | (5) | (6) |

Element Type and Size Constraints

(1) advertisement 70 10, background image 90 144, image 21 7, text 90 25, text 90 8, text button 54 10, text button 54 10, text button 8 7, text button 33 7, text button 54 10, web view 70 10, web view 70 10

(2) icon 12 12, text 29 5, text button 85 10, text button 85 10, toolbar 90 12

(3) input 90 110, text 48 8, text 38 8, text button 28 9, text button 28 9, toolbar 90 16

(4) icon 7 7, image 69 39, text 90 5, text 90 6, text button 83 38

(5) input 53 11, on/off switch 85 11, radio button 17 11, radio button 19 11, text 15 13, text 74 5, text 26 4, text button 15 9, text button 15 8, text button 45 7, text button 45 7

(6) image 71 60, text 81 9, text 76 26, text button 72 16

(a) RICO

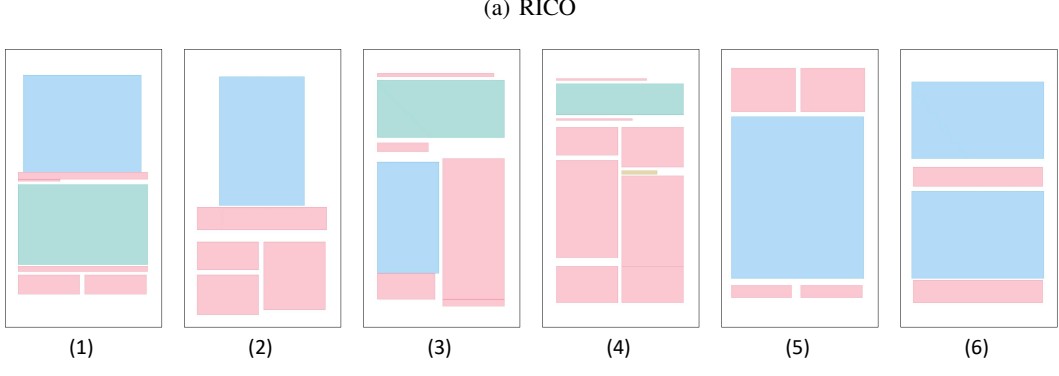

| | | | | | |
|---|---|---|---|---|---|
| (1) | (2) | (3) | (4) | (5) | (6) |

Element Type and Size Constraints

(1) figure 90 56, table 99 46, text 99 4, text 32 1, text 99 3, text 47 11, text 47 11

(2) figure 65 74, text 99 13, text 47 16, text 47 39, text 47 23

(3) figure 47 64, table 97 33, text 89 2, text 39 5, text 47 81, text 44 15, text 47 4

(4) table 97 18, text 69 1, text 58 1, text 47 23, text 47 16, text 47 56, text 47 52, text 47 21, text 47 21, title 27 2

(5) figure 101 93, text 49 25, text 49 25, text 46 7, text 47 7

(6) figure 101 44, figure 101 50, text 99 11, text 99 13

(b) PubLayNet

Figure 10: Qualitative results of Gen-TS on RICO and PubLayNet. The element type and size constraints are in the table.

## C.3 Generation Conditioned on Element Relationships

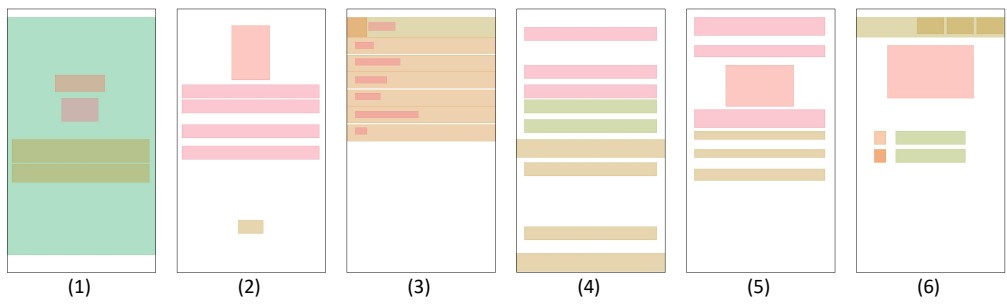

| | | | | | |
|---|---|---|---|---|---|
| (1) | (2) | (3) | (4) | (5) | (6) |

Element Relationship Constraints

(1) background image, image, text, text button*2, text 2 center canvas, text button 3 center canvas, text 2 smaller background image 0

(2) image, text*4, text button, text 4 bottom image 0, text 3 larger text 2, text button 5 bottom text 4

(3) icon, list item*6, text*7, toolbar, list item 2 top canvas, list item 4 center canvas, list item 6 center canvas, list item 1 bottom icon 0, list item 2 larger icon 0, list item 4 larger icon 0, text 7 right icon 0, toolbar 14 top list item 1, list item 3 equal list item 2, list item 5 bottom list item 2, text 7 top list item 2, text 13 smaller list item 2, text 7 smaller list item 3

(4) input*2, text*3, text button*4, text button 7 bottom canvas, text 2 equal input 0, text 4 top input 0, text button 6 smaller input 0, text 3 smaller input 1, text button 7 smaller input 1, text button 7 bottom input 1, text button 6 larger text 3, text button 6 bottom text 4

(5) image, text*3, text button*3, text 2 top canvas, text 3 bottom image 0, text 3 bottom text 2, text button 5 bottom text 2, text button 5 top text 3

(6) icon*3, image, input*2, text button*3, toolbar, input 5 center canvas, text button 6 center canvas, icon 2 equal icon 1, text button 6 larger icon 1, text button 8 larger icon 1, input 4 bottom image 3, text button 8 smaller image 3, text button 7 bottom input 4, text button 8 larger input 5, text button 8 bottom input 5, text button 8 larger text button 6

(a) RICO

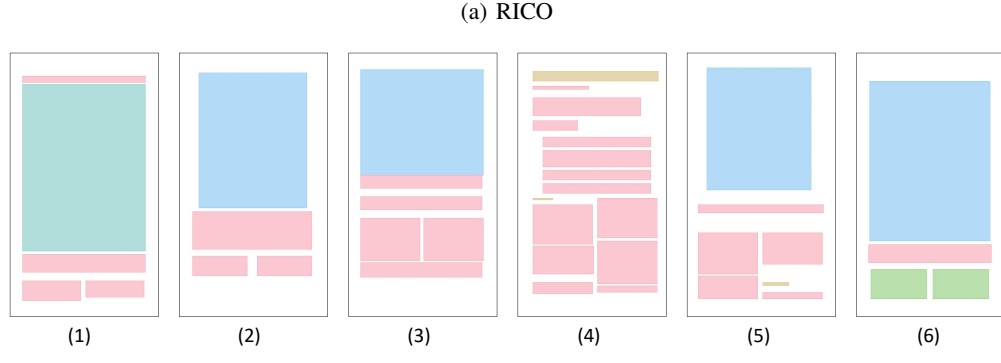

| | | | | | |
|---|---|---|---|---|---|
| (1) | (2) | (3) | (4) | (5) | (6) |

Element Relationship Constraints

(1) table, text*4, text 4 bottom text 1, text 4 smaller text 3

(2) figure, text*3, text 2 smaller figure 0, text 2 bottom figure 0

(3) figure, text*5, text 1 center canvas, text 5 bottom canvas, text 5 bottom text 2

(4) text*13, title*2, title 13 top canvas, text 4 bottom text 0, text 2 bottom text 1, text 4 bottom text 1, text 6 smaller text 1, text 9 larger text 1, title 13 top text 1, title 14 smaller text 1, title 13 smaller text 2, text 8 larger text 3, text 9 bottom text 3, text 6 bottom text 4, text 8 bottom text 4, text 10 larger text 4, text 9 bottom text 6, text 12 larger text 6, title 13 larger text 6, text 10 larger text 8, text 12 smaller text 9, text 12 smaller text 10, title 13 smaller text 10, title 14 top text 10, text 12 smaller text 11

(5) figure, text*5, title, text 5 smaller figure 0, text 5 bottom figure 0, text 5 smaller text 2, title 6 bottom text 3, title 6 center text 5

(6) figure, list*2, text, text 3 bottom figure 0, list 2 right list 1

(b) PubLayNet

Figure 11: Qualitative results of Gen-R on RICO and PubLayNet. The element relationship constraints are in the table.

## C.4 Layout Completion

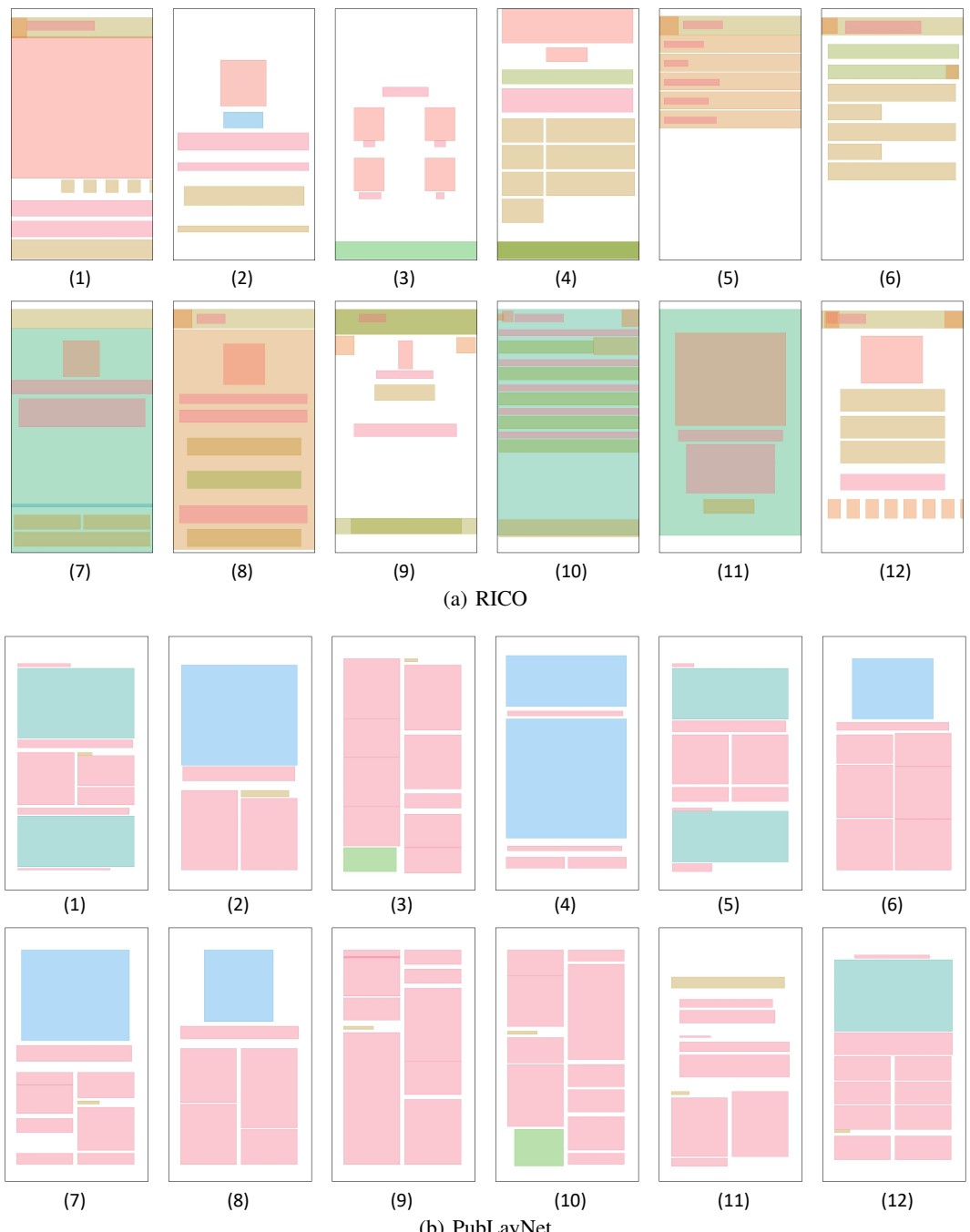

Figure 12: Qualitative results of completion on RICO and PubLayNet.

## C.5 Layout Refinement

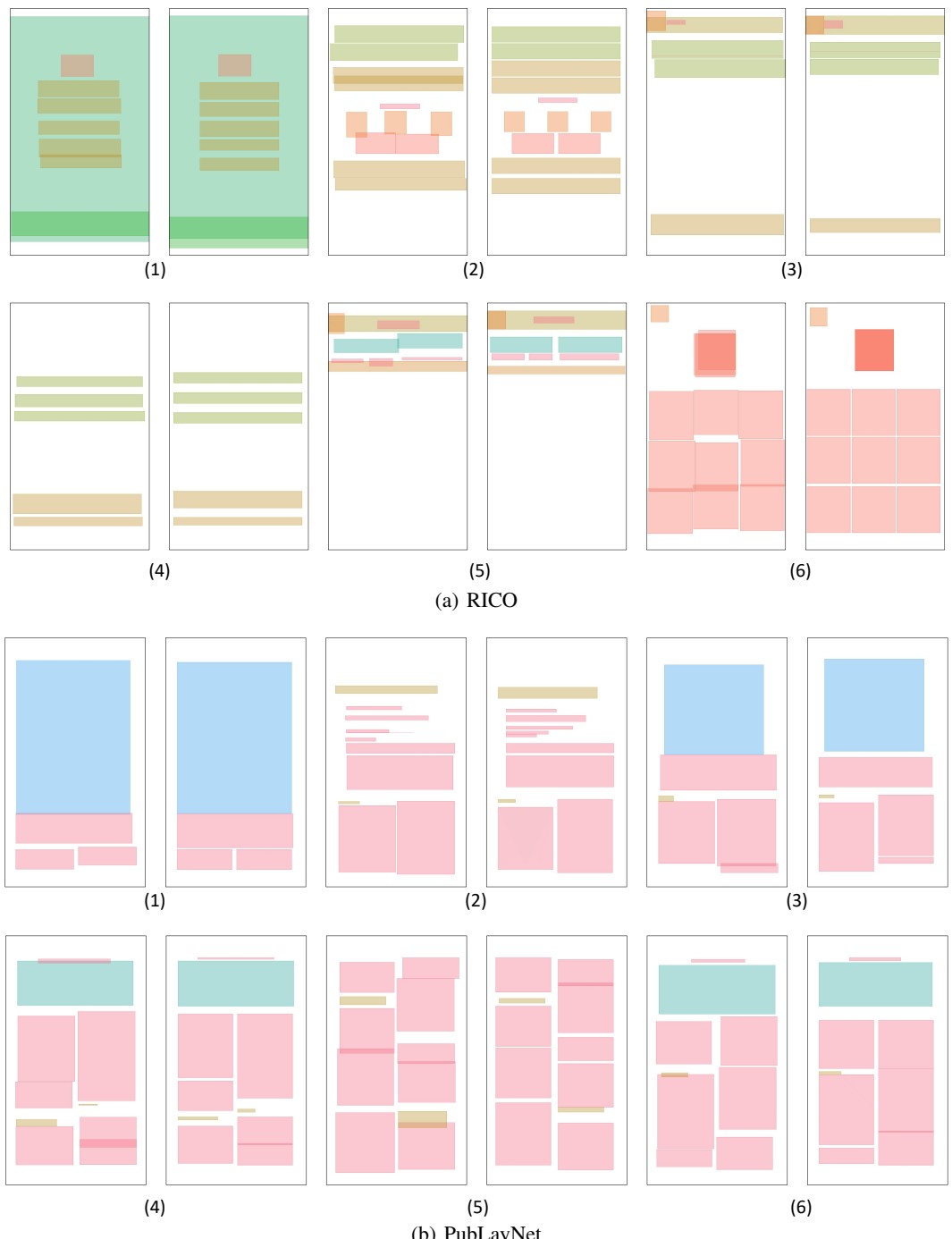

(1)

(2)

(3)

(4)

(5)

(6)

(a) RICO

(1)

(2)

(3)

(4)

(5)

(6)

(b) PubLayNet

Figure 13: Qualitative results of refinement on RICO and PubLayNet. Note that each group has two layouts. The left one is the noisy layout, and the right one is the refined layout.

## C.6    Content-Aware Layout Generation

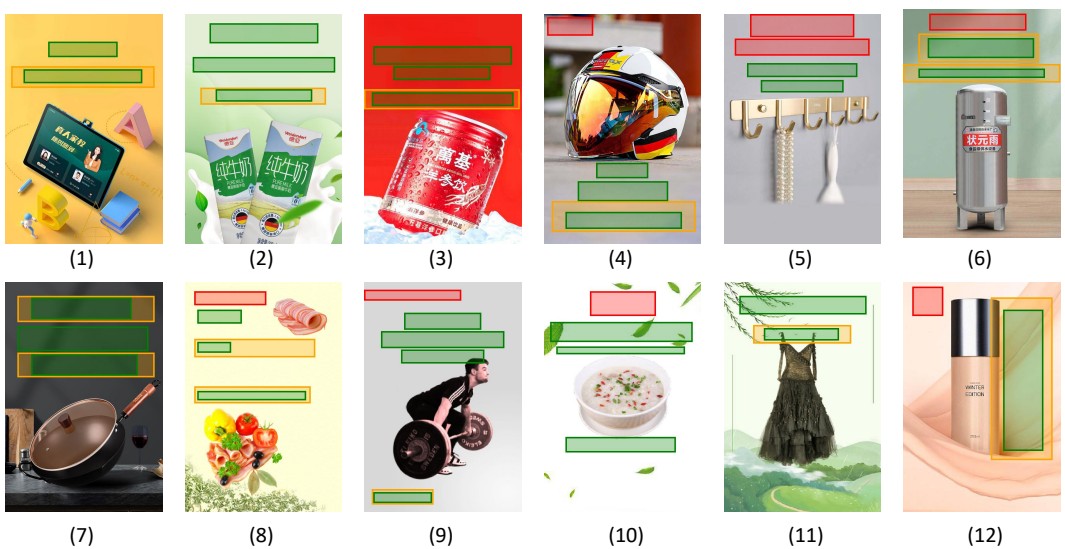

Figure 14: Qualitative results of content-aware layout generation on PosterLayout.

## C.7    Text-to-Layout

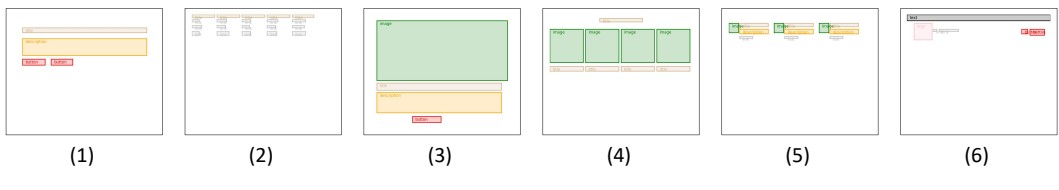

| Textual Descriptions |
|---|
| (1) A page to file an insurance claim. There should be two text boxes. One is for the title 'Need to file an insurance claim' and the other is a short description. Two buttons are necessary to click to file or manage the claim. |
| (2) A footer page showing the homepage of a company. There should be five entries of text links. And the text links should be unfolded. Users can get access to different info through these links. |
| (3) A page to introduce an online editor. The page should contain an image, and the slogan "Online Infographics, Diagrams, & Charts Maker" to have a brief introduction, and a further description of the specific service of the editor. Then a button is needed for the user to click to learn info. |
| (4) A page for introducing the home tours. The page should have a title, "home tours". Also, there are four groups and each group has an image and a title. |
| (5) A page for introducing the service provided by a website. The page should have three groups of information sets: "Web Development" "Social Media", and "Link Building". Each group should include a small image, a title, a further introduction, and a link "READ MORE" for more info by clicking it. |
| (6) A header page for navigation of an online shop. The page should have an announcement to explain the product is out of stock, a logo, and two links "HOME" and "BUY DITTO HANGERS" for the user to return to the homepage and view products to shop. Also, two buttons for search and shopping cart are needed. |

Figure 15: Qualitative results of text-to-layout on WebUI. The textual descriptions are in the table.

# D Prompt Examples

Here, we show a prompt example for each layout generation task, including Gen-T (Table 11), Gen-TS (Table 12), Gen-R (Table 13), completion (Table 14), refinement (Table 15), content-aware layout generation (Table 16), and text-to-layout (Table 17).

---

**Preamble**
Please generate a layout based on the given information.
*Task Description*: generation conditioned on given element types
*Layout Domain*: android layout
*Canvas Size*: canvas width is 90px, canvas height is 160px

**Exemplar 1**
*Element Type Constraint*: image | image | image | image | text | text | text button
```
<html>
<body>
<div class="image" style="left:15px; top:42px; width:51px; height:82px"></div>
<div class="image" style="left:52px; top:77px; width:22px; height:50px"></div>
<div class="image" style="left:54px; top:96px; width:18px; height:14px"></div>
<div class="image" style="left:0px; top:143px; width:90px; height:1px"></div>
<div class="text" style="left:34px; top:15px; width:21px; height:7px"></div>
<div class="text" style="left:22px; top:22px; width:44px; height:9px"></div>
<div class="text button" style="left:2px; top:147px; width:41px; height:10px"></div>
</body>
</html>
```

**Exemplar 2**
*Element Type Constraint*: image | image | image | image | pager indicator | text | text | text button | text button | text button
```
<html>
<body>
<div class="image" style="left:0px; top:5px; width:90px; height:93px"></div>
<div class="image" style="left:30px; top:8px; width:29px; height:10px"></div>
<div class="image" style="left:38px; top:86px; width:12px; height:12px"></div>
<div class="image" style="left:32px; top:86px; width:24px; height:12px"></div>
<div class="pager indicator" style="left:0px; top:119px; width:90px; height:5px"></div>
<div class="text" style="left:0px; top:98px; width:90px; height:10px"></div>
<div class="text" style="left:0px; top:109px; width:90px; height:9px"></div>
<div class="text button" style="left:0px; top:124px; width:45px; height:10px"></div>
<div class="text button" style="left:45px; top:124px; width:45px; height:10px"></div>
<div class="text button" style="left:0px; top:138px; width:90px; height:11px"></div>
</body>
</html>
```

......

**Test Sample**
*Element Type Constraint*: icon | image | image | text | text | text button | toolbar
(Generated by LLMs)

---

Table 11: A prompt example of Gen-T on RICO.

**Preamble**
Please generate a layout based on the given information.
*Task Description*: generation conditioned on given element types and sizes
*Layout Domain*: android layout
*Canvas Size*: canvas width is 90px, canvas height is 160px

**Exemplar 1**
*Element Type and Size Constraint*: icon 12 12 | image 0 0 | input 81 10 | input 81 10 | text 81 4 | text 22 4 | text button 85 10 | text button 10 4 | toolbar 90 12

```
<html>
<body>
<div class="icon" style="left:0px; top:5px; width:12px; height:12px"></div>
<div class="image" style="left:12px; top:11px; width:0px; height:0px"></div>
<div class="input" style="left:4px; top:40px; width:81px; height:10px"></div>
<div class="input" style="left:4px; top:28px; width:81px; height:10px"></div>
<div class="text" style="left:4px; top:23px; width:81px; height:4px"></div>
<div class="text" style="left:13px; top:9px; width:22px; height:4px"></div>
<div class="text button" style="left:2px; top:62px; width:85px; height:10px"></div>
<div class="text button" style="left:75px; top:43px; width:10px; height:4px"></div>
<div class="toolbar" style="left:0px; top:5px; width:90px; height:12px"></div>
</body>
</html>
```

**Exemplar 2**
*Element Type and Size Constraint*: card 86 41 | icon 12 12 | input 64 12 | input 78 12 | input 78 9 | input 61 9 | text 15 5 | text button 25 7 | text button 13 7 | text button 62 3 | toolbar 90 12

```
<html>
<body>
<div class="card" style="left:1px; top:19px; width:86px; height:41px"></div>
<div class="icon" style="left:0px; top:5px; width:12px; height:12px"></div>
<div class="input" style="left:5px; top:36px; width:64px; height:12px"></div>
<div class="input" style="left:5px; top:23px; width:78px; height:12px"></div>
<div class="input" style="left:5px; top:23px; width:78px; height:9px"></div>
<div class="input" style="left:5px; top:36px; width:61px; height:9px"></div>
<div class="text" style="left:15px; top:8px; width:15px; height:5px"></div>
<div class="text button" style="left:60px; top:51px; width:25px; height:7px"></div>
<div class="text button" style="left:70px; top:38px; width:13px; height:7px"></div>
<div class="text button" style="left:13px; top:62px; width:62px; height:3px"></div>
<div class="toolbar" style="left:0px; top:5px; width:90px; height:12px"></div>
</body>
</html>
```

......

**Test Sample**
*Element Type and Size Constraint*: icon 12 12 | input 83 9 | input 83 9 | text 83 8 | text button 19 9 | text button 77 5 | toolbar 90 12
(Generated by LLMs)

Table 12: A prompt example of Gen-TS on RICO.

Please generate a layout based on the given information.
*Task Description*: generation conditioned on given element relationships
*Layout Domain*: android layout
*Canvas Size*: canvas width is 90px, canvas height is 160px

**Exemplar 1**
*Element Type Constraint*: image | image | image | text | text | text | text | text button | toolbar
*Element Relationship Constraint*: text 5 bottom canvas | image 1 larger image 0 | text 3 larger image 0 | text 5 larger image 0 | toolbar 8 larger image 0 | image 2 equal image 1 | text 4 smaller image 2 | text 6 smaller image 2 | toolbar 8 top text 4

```
<html>
<body>
<div class="image" style="left:0px; top:7px; width:7px; height:7px"></div>
<div class="image" style="left:31px; top:33px; width:28px; height:29px"></div>
<div class="image" style="left:30px; top:101px; width:28px; height:29px"></div>
<div class="text" style="left:8px; top:8px; width:28px; height:5px"></div>
<div class="text" style="left:24px; top:66px; width:40px; height:5px"></div>
<div class="text" style="left:18px; top:133px; width:52px; height:5px"></div>
<div class="text" style="left:18px; top:140px; width:51px; height:7px"></div>
<div class="text button" style="left:75px; top:5px; width:14px; height:11px"></div>
<div class="toolbar" style="left:0px; top:5px; width:90px; height:11px"></div>
</body>
</html>
```

**Exemplar 2**
*Element Type Constraint*: text | text | text | text | text button
*Element Relationship Constraint*: text 3 bottom text 0 | text 2 equal text 1

```
<html>
<body>
<div class="text" style="left:0px; top:7px; width:90px; height:5px"></div>
<div class="text" style="left:3px; top:19px; width:83px; height:30px"></div>
<div class="text" style="left:3px; top:57px; width:83px; height:30px"></div>
<div class="text" style="left:3px; top:95px; width:83px; height:52px"></div>
<div class="text button" style="left:0px; top:148px; width:90px; height:11px"></div>
</body>
</html>
```

......

**Test Sample**
*Element Type Constraint*: icon | image | text | text | text | text | text button | text button
*Element Relationship Constraint*: text 3 top canvas | text 5 top canvas | text 2 right icon 0 | text button 6 bottom icon 0 | text 3 bottom image 1 | text button 7 bottom text 4
(Generated by LLMs)

Table 13: A prompt example of Gen-R on RICO.

Please generate a layout based on the given information.

*Task Description*: layout completion

*Layout Domain*: android layout

*Canvas Size*: canvas width is 90px, canvas height is 160px

**Exemplar 1**

*Partial Layout*: image 21 5 47 40

```
<html>
<body>
<div class="image" style="left:21px; top:5px; width:47px; height:40px"></div>
<div class="text button" style="left:2px; top:53px; width:84px; height:15px"></div>
<div class="image" style="left:7px; top:74px; width:9px; height:5px"></div>
<div class="text" style="left:19px; top:74px; width:67px; height:5px"></div>
<div class="text button" style="left:2px; top:85px; width:84px; height:14px"></div>
<div class="text button" style="left:1px; top:104px; width:86px; height:12px"></div>
<div class="text button" style="left:1px; top:136px; width:86px; height:11px"></div>
</body>
</html>
```

**Exemplar 2**

*Partial Layout*: image 17 5 56 11

```
<html>
<body>
<div class="image" style="left:17px; top:5px; width:56px; height:11px"></div>
<div class="image" style="left:0px; top:17px; width:90px; height:48px"></div>
<div class="text" style="left:2px; top:65px; width:86px; height:48px"></div>
<div class="image" style="left:0px; top:108px; width:90px; height:5px"></div>
<div class="pager indicator" style="left:38px; top:114px; width:12px; height:8px"></div>
<div class="text button" style="left:3px; top:124px; width:82px; height:13px"></div>
<div class="text button" style="left:62px; top:137px; width:17px; height:10px"></div>
<div class="text" style="left:10px; top:140px; width:51px; height:6px"></div>
</body>
</html>
```

......

**Test Sample**

*Partial Layout*: image 12 10 65 32

(Generated by LLMs)

Table 14: A prompt example of layout completion on RICO.

**Preamble**

Please generate a layout based on the given information.

***Task Description***: layout refinement

***Layout Domain***: android layout

***Canvas Size***: canvas width is 90px, canvas height is 160px

**Exemplar 1**

***Noise Layout***: advertisement 11 18 70 11 | icon 76 5 11 11 | icon 0 6 12 10 | image 16 8 13 11 | text 30 3 21 5 |
text 29 11 23 4 | toolbar 0 5 88 16 | web view 9 16 69 12 | web view 11 17 70 12 | web view 0 20 90 140

```
<html>
<body>
<div class="advertisement" style="left:10px; top:18px; width:70px; height:11px"></div>
<div class="icon" style="left:77px; top:6px; width:12px; height:11px"></div>
<div class="icon" style="left:0px; top:5px; width:12px; height:13px"></div>
<div class="image" style="left:15px; top:6px; width:14px; height:11px"></div>
<div class="text" style="left:30px; top:6px; width:21px; height:6px"></div>
<div class="text" style="left:30px; top:12px; width:23px; height:5px"></div>
<div class="toolbar" style="left:0px; top:5px; width:90px; height:13px"></div>
<div class="web view" style="left:10px; top:18px; width:70px; height:11px"></div>
<div class="web view" style="left:10px; top:18px; width:70px; height:11px"></div>
<div class="web view" style="left:0px; top:18px; width:90px; height:141px"></div>
</body>
</html>
```

**Exemplar 2**

***Noise Layout***: advertisement 0 4 89 11 | background image 0 4 89 145 | icon 4 17 6 7 | icon 11 19 4 6 | image 1 8 5 5 |
image 0 13 20 10 | text 35 8 18 5 | text button 80 6 7 3 | text button 16 14 64 8 | text button 82 14 9 7 |
text button 10 29 68 11 | text button 0 39 88 12 | web view 10 2 69 12 | web view 9 6 69 10

```
<html>
<body>
<div class="advertisement" style="left:0px; top:5px; width:90px; height:10px"></div>
<div class="background image" style="left:0px; top:5px; width:90px; height:144px"></div>
<div class="icon" style="left:5px; top:19px; width:4px; height:4px"></div>
<div class="icon" style="left:11px; top:19px; width:4px; height:4px"></div>
<div class="image" style="left:2px; top:7px; width:5px; height:5px"></div>
<div class="image" style="left:0px; top:16px; width:21px; height:7px"></div>
<div class="text" style="left:35px; top:7px; width:18px; height:5px"></div>
<div class="text button" style="left:81px; top:8px; width:6px; height:5px"></div>
<div class="text button" style="left:16px; top:16px; width:63px; height:10px"></div>
<div class="text button" style="left:81px; top:16px; width:8px; height:7px"></div>
<div class="text button" style="left:11px; top:27px; width:68px; height:10px"></div>
<div class="text button" style="left:0px; top:41px; width:90px; height:11px"></div>
<div class="web view" style="left:10px; top:5px; width:70px; height:10px"></div>
<div class="web view" style="left:10px; top:5px; width:70px; height:10px"></div>
</body>
</html>
```

......

**Test Sample**

***Noise Layout***: icon 68 5 10 12 | icon 1 5 9 12 | icon 80 5 12 13 | text 14 7 56 2 | toolbar 0 5 90 10 |
web view 0 18 90 130 | web view 0 19 90 130

(Generated by LLMs)

Table 15: A prompt example of layout refinement on RICO.

**Preamble**

Please generate a layout based on the given information.

*Task Description*: content-aware layout generation

*Layout Domain*: poster layout

*Canvas Size*: canvas width is 102px, canvas height is 150px

**Exemplar 1**

*Content Constraint*: left 25px, top 25px, width 30px, height 12px

*Element Type Constraint*: logo | text | underlay | text | text

```
<html>
<body>
<div class="logo" style="left:34px; top:14px; width:66px; height:23px"></div>
<div class="text" style="left:10px; top:25px; width:94px; height:36px"></div>
<div class="underlay" style="left:18px; top:37px; width:85px; height:48px"></div>
<div class="text" style="left:36px; top:40px; width:64px; height:45px"></div>
<div class="text" style="left:28px; top:48px; width:74px; height:53px"></div>
</body>
</html>
```

**Exemplar 2**

*Content Constraint*: left 23px, top 60px, width 56px, height 69px

*Element Type Constraint*: logo | text | underlay | text

```
<html>
<body>
<div class="logo" style="left:35px; top:0px; width:66px; height:10px"></div>
<div class="text" style="left:15px; top:22px; width:86px; height:33px"></div>
<div class="underlay" style="left:29px; top:37px; width:73px; height:49px"></div>
<div class="text" style="left:35px; top:40px; width:67px; height:47px"></div>
</body>
</html>
```

......

**Test Sample**

*Content Constraint*: left 26px, top 62px, width 50px, height 60px

*Element Type Constraint*: logo | text | text | text | underlay | text

(Generated by LLMs)

Table 16: A prompt example of content-aware layout generation on PosterLayout.

**Preamble**

Please generate a layout based on the given information.

*Task Description*: text-to-layout

*Layout Domain*: web layout

*Canvas Size*: canvas width is 120px, canvas height is 120px

**Exemplar 1**

*Text*: A header page for the company Fashionably Latellc. On the page, there should include
a name of the company, three navigation links "Home" "Our Fashionably Items" and "Return/Refund Policy" for the user
to click to return to the homepage, view items of the company, and know the return or refund policy. Besides, it is necessary
to have three icons for login, search, and shopping cart.

```
<html>
<body>
<div class="icon" style="left:101px; top:3px; width:2px; height:2px"></div>
<div class="icon" style="left:106px; top:3px; width:2px; height:2px"></div>
<div class="icon" style="left:110px; top:2px; width:5px; height:5px"></div>
<div class="link" style="left:35px; top:3px; width:4px; height:2px"></div>
<div class="link" style="left:42px; top:3px; width:17px; height:2px"></div>
<div class="link" style="left:62px; top:3px; width:16px; height:2px"></div>
<div class="title" style="left:5px; top:3px; width:24px; height:3px"></div>
</body>
</html>
```

**Exemplar 2**

*Text*: A header page for a website Png AAA. There should have a logo on the left, and three
links "LOG IN" "SIGN UP", and "UPLOAD", so the user can click them to log in, create an account, and upload something.

```
<html>
<body>
<div class="link" style="left:88px; top:2px; width:5px; height:2px"></div>
<div class="link" style="left:97px; top:2px; width:6px; height:2px"></div>
<div class="link" style="left:110px; top:2px; width:6px; height:2px"></div>
<div class="logo" style="left:2px; top:1px; width:15px; height:4px"></div>
</body>
</html>
```

......

**Test Sample**

*Text*: A header page of the website "homment". On the page, there should include a logo
of the website. Five links ("Latest", "Top100", "About", "Register", and "Login") a button "Create" and an icon are on the page.
(Generated by LLMs)

Table 17: A prompt example of text-to-layout on WebUI.