# OpenReview forum: "LayoutPrompter: Awaken the Design Ability of Large Language Models"
_NeurIPS.cc/2023/Conference — NeurIPS 2023 poster_

### Official Review · Reviewer_Lp5t · 2023-06-30

**Soundness:** 2 fair
**Presentation:** 3 good
**Contribution:** 2 fair
**Rating:** 6
**Confidence:** 4

**Summary:**

This paper presents a set of few-shot techniques to enable large language models to generate geometric layouts, particularly in the domain of User interfaces, posters and documents. These techniques include dynamic prompting and layout ranker modules, which allows the LLMs to learn from relevant examples in the dataset and filter out badly generated UIs from various samples. The presented technique enables cross-domain, task-generic generation of layouts without the need for training, utilizing existing knowledge in LLMs which was unexplored in prior work.


**Strengths:**

- Enabled LLMs to perform a novel task. This task is also typically challenging for LLMs to perform due to its highly geometric nature.

- Experiments and prompts that lead to the success of layout generation in LLMs would constitute a significant contribution, given the challenging nature of this task in LLM.

- Introduced dynamic prompting and layout ranker, of which their formulation and implementation can be useful for future layout generation work

- Achieved competitive performance in some layout generation tasks, without the need for training


**Weaknesses:**

Some of these weaknesses are pointed out in the limitations sections by the authors themselves, but I would like to highlight them since I believe they are important for the significance of the current paper:

- **Reliant on existing dataset's coverage of the target task to achieve good performance:** This is shown in the ablation study that the FID is significantly higher without dynamic prompting. This is an important issue in the UI domain where data is difficult and costly to collect.

- **Generalisation ability of LLMs in generating UIs:** Since the dynamic prompting module relies heavily on the provided prompt, it is unclear whether the generated UIs are novel and are meaningful variations for design inspiration. This can be shown by computing the similarity between the generated UIs and the prompt UIs, and comparing this with existing work (which might be generated UIs against the entire dataset for existing techniques that don't use prompts).

- **Diversity and Coverage of the generated UIs:** Related to the above point, it is unclear how much of the entire design space that the generated UIs can cover. Existing work computes the Wasserstein distance between that of the generated UIs and the test set, of which this work did not compute (partially due to the conditional nature of all evaluated tasks). I wonder if the author can perform some unconditional UI generation experiments to explore the knowledge of the overall UI design space that is learned by existing LLMs.


**Questions:**

These questions are directly relevant to the weaknesses of the paper:

1) What is the average qualitative and quantitative similarity of the generated UIs against the most similar UIs in the prompt? It would be great if the author(s) can use an established metric (e.g., DocSim) and compare this similarity against existing work. Existing work that doesn't use prompt can be compared against the most similar entry in the entire dataset.

2) What is the pool of UIs that the prompt UIs can be sourced from? Is this only from the training set of various domains?

3) How does the performance of the model correlate with the size of the dataset that the prompt UIs can be sourced from? It would be great if the author(s) can perform some ablation experiments along this dimension.

4) Can the author(s) discuss further the significant drop of FIDs w/o dynamic prompting? This discussion could be qualitative.


**Limitations:**

Limitations are adequately addressed.

---

> ### Author Rebuttal · Authors · 2023-08-10
>
> **Weakness1. Reliant on existing dataset's coverage of the target task to achieve good performance.**
>
> To understand the effect of dataset's coverage on model performance, we regard the size of the layout pool that prompt UIs can be sourced from as a proxy for coverage, and we study how the model performance on Gen-T task varies as the pool size increases.
> Being mindful of the LLM experiment costs and the limited time for rebuttal, we randomly select 1,000 samples from the test set of RICO and PubLayNet for the experiment.
> The results are reported in Table 15 (in the attachment).
> It is observed that when the pool size is only 100 (i.e., small coverage), all the metrics are poor on both datasets.
> However, when the pool size increases to 500, all the metrics show significant improvement, in particular FID.
> When the pool size continues to increase from 500 to 10,000, although the overall trend of FID is still improving, the change speed slows down, and the other metrics only have marginal improvement.
> With a size of 10,000 layout pool, the model has already been on par with the one using the full training set (Full Set).
> To sum up, although increasing the coverage of the dataset can indeed improve the model performance, the improvement is limited when the coverage is increased to a certain extent.
> In the revised version, We will experiment with the full test set on more tasks and add the results in the supplementary material.
>
>
> **Weakness2. Generalization ability of LLMs in generating UIs.**
>
> To investigate the generalization ability of LayoutPrompter, we take your kind suggestion and compute the similarity (measured by $DocSim_g$) between the generated UIs and prompt UIs.
> We also compare with LayoutFormer++, for which the $DocSim_g$ is computed between the generated layouts and training layouts.
> We present the quantitative and qualitative results in Table 13 and Figure 16 (in the attachment).
> Quantitative results show that LayoutPrompter achieves a competitive or even better $DocSim_g$ score with LayoutFormer++, indicating that LayoutPrompter has a close generalization ability to LayoutFormer++.
> Qualitative results indicate that LayoutPrompter is capable of generating meaningful variations different from the retrieved layouts.
>
> **Weakness3. Diversity and Coverage of the generated UIs.**
>
> Thanks for your advice.
> We do think our approach can perform unconditional layout generation tasks.
> To achieve so, we should randomly choose prompt layouts in Dynamic Prompting rather than using ConsSim, as there are no constraints as input for prompt layouts retrieval.
> Our preliminary hypothesis for unconditional layout generation is that with the LLM prior design knowledge and a properly sized layout pool, LayoutPrompter could perform on par or even better performance with state-of-the-art layout approaches.
> We will implement this and include a fair experiment in the revised version to comprehensively investigate the problem.
>
> In addition, FID is a measure of the distribution difference between the generated layouts and the real layouts.
> Therefore, we argue that the competitive FID of our method on the representative conditional layout generation tasks could also reflect the diversity and coverage of the generated layouts.
>
> **Question1. The qualitative and quantitative similarity of the generated UIs against the most similar UIs in the prompt.**
>
> Please refer to our response in Weakness2 for the study results.
>
> **Question2. What is the pool of UIs that the prompt UIs can be sourced from?**
>
> The prompt UIs are all sourced from the training sets of various domains.
> We follow the dataset splits in the baselines for fair comparisons.
> Specifically, the splits of RICO and PubLayNet follow LayoutFormer++, and the splits of PosterLayout follow DS-GAN.
>
> **Question3. How does the performance of the model correlate with the size of the dataset that the prompt UIs can be sourced from?**
>
> Please refer to our response in Weakness1 for the study results.
>
>
>
> **Question4. The significant drop of FIDs w/o dynamic prompting.**
>
> Dynamic Prompting is responsible for selecting prompt layouts from the training set with constraints similar to a given one.
> It helps the model understand the characteristics of various layout constraints and thus generate high-quality layouts conforming to the given constraint.
> Without Dynamic Prompting, the model hardly acquires the layout patterns for certain constraints, leading to an inferior FID.
> For example, the RICO dataset contains a ``drawer'' element, which represents the side navigation bar and is usually placed on the left side of the canvas.
> If the model is not familiar with its meaning and layout pattern, it is likely that the drawer will not be placed in the correct position, and thus the generated layout will not be a plausible one, ultimately hurting the FID score.
> We also include more qualitative results for the variant w/o Dynamic Prompting in Figure 19 (in the attachment).

---

> > ### Comment · Reviewer_Lp5t · 2023-08-16
> >
> > Thanks for addressing my concerns and adding extensive additional experiments during the rebuttal period!
> >
> > Most of my concerns have been adequately addressed, and I increased my score from 4 to 6. I encourage the author(s) to consider discussing the weaknesses raised in this discussion in the revised paper.

---

### Official Review · Reviewer_mVCb · 2023-07-01

**Soundness:** 4 excellent
**Presentation:** 3 good
**Contribution:** 3 good
**Rating:** 7
**Confidence:** 3

**Summary:**

The authors propose LayoutPrompter, which is a new approach for generating high-quality, cross-domain graphic layouts without any model training or fine-tuning. It can handle a variety of conditional layout generation tasks and outperforms existing methods on some metrics. The approach is competitive with state-of-the-art approaches on five traditional conditional layout generation tasks and even extends to solve two challenging problems that have more flexible constraints.

**Strengths:**

The strength of this submission lies in its clear motivation, well-written content, and ease of understanding. The authors have effectively articulated the purpose and significance of their research, making the motivation behind the study readily apparent to the reader. The paper is written in a coherent and structured manner, which contributes to its ease of follow throughout. Furthermore, the study impressively conducts solid experiments that encompass a range of layout design settings, providing robust empirical evidence to support their findings. Overall, the submission showcases a strong foundation and execution, making it a valuable contribution to the field.

**Weaknesses:**

(1) One major weakness of the submission is the lack of detailed information regarding the new dataset, WebUI. While the authors mention its collection, they provide very limited information about the data collection process or the format of the dataset. This omission raises several questions that remain unanswered. It would be beneficial for the authors to include a section in the main paper or supplementary material that thoroughly describes the data collection process, including the sources, selection criteria, and any preprocessing steps applied. Additionally, specifying the format of the dataset (e.g., file type, structure) would provide valuable insights for readers.

(2) Another weakness pertains to the organization and clarity of the dataset setup explanation. As the submission covers multiple tasks and settings, it would greatly enhance readability if the authors were more specific about which dataset was used for each setting. Currently, the explanation of the dataset setup (lines 227-236) appears disorganized and challenging to follow. (I searched for the experimental results on WebUI and found nothing, then it took me a while to figure out that WebUI was referred to as text-to-layout…) To address this, I recommend the authors consider incorporating a table that lists the tasks and corresponding datasets. This table would help readers quickly identify the datasets utilized for each specific setting, reducing confusion and facilitating a more straightforward interpretation of the experimental results.


**Questions:**

(1) Figure 1 is a bit hard to understand as a teaser figure. It took me a while to figure out that the left parts refer to the “ five typical tasks and two more challenging tasks”, while the right parts are the subdomains of discussion…

(2) Missing reference for saliency map on line 161

(3) Are there any ablation studies on the effect of the number of prompting examples?

(4) Regarding the new dataset:
- What is the source of the website?
- What kind of HTML elements are included?
- What is the portion of each element?
- How was the quality controlled (e.g., removing ads, removing unrelated banner, etc)?
- How was the instructions for layout design collected?
- Does the instruction creation process involve human effort or was it automatically generated?
- Where is the data sheet for this new dataset?

(5) Out of curiosity, may I ask why the authors did not conduct any human evaluation?


**Limitations:**

The authors have included discussions of the limitations.

---

> ### Author Rebuttal · Authors · 2023-08-10
>
> **Weakness1. The details of WebUI dataset are missing.**
>
> Thanks for your advice. We'll include a section in the supplementary material to describe the dataset details.
> Here, we first provide you with the information.
>
> 1) Source.
> We mainly collect the company homepages of the Fortune 500 companies (Microsoft, Google and so on).
> Since these web pages are very long, we divide them into several sections according to the id values in the HTML (Header section, Hero section, Product section, Footer section and so on).
> We then filter ads and cookies based on text content to control the dataset quality.
>
> 2) From HTML to layouts.
> In the last step, we have acquired many web sections.
> We convert these web sections into corresponding layouts in this step.
> Since HTML non-leaf nodes are usually used as containers without specific semantics, we only include leaf nodes in the final layouts and filter out all non-leaf nodes.
> Then, we determine the element type by the HTML tag.
> For example, elements within an "h" tag are considered "title", elements within an "a" tag are considered "link", elements within a "div" tag are considered "text", elements with a "src" attribute are treated as "image" and so on.
> Element positions can be automatically extracted from the "top", "left", "width" and "height" attributes.
> So far, we have converted HTML into the corresponding layouts.
> We filter layouts with more than 50 elements. We also filter out heavily overlapping layouts.
>
> 3) Text annotation.
> When the layouts are ready, we recruit annotators to annotate natural language descriptions for each layout.
> First, annotators were asked to label 30 samples. Then, one expert checked their results and gave them feedback. This training was performed in 2 rounds to select 15 qualified annotators.
> After annotation, 5 experts evaluated each data sample. A
> sample was accepted only when more than 3 experts agreed
> with it.
> Finally, we got about 5000 text-layout pairs to construct the first text-to-layout dataset, WebUI.
>
> 4) Data sheet.
> Each sample in our constructed dataset is a text-layout pair.
> And each layout contains $N$ element types and $N\times4$ coordinate attributes.
> WebUI has a total of 5212 text-layout pairs.
> The average text length is 40.5.
> The average number of elements is 10.1.
> The maximum number of elements is 50.
>
> **Weakness2. The dataset setup is disorganized.**
>
> Thanks for your advice.
> We'll add a table as you suggested in the revised version for better readability.
> Here, we clarify that RICO and PubLayNet are used in 5 typical layout generation tasks, PosterLayout is used in content-aware layout generation, and our constructed WebUI is used in text-to-layout.
>
> **Question1. The teaser figure is hard to understand.**
>
> We will revise the figure title, explaining the meaning of each part of the teaser figure, to make it easier for readers to understand.
>
> **Question2. Missing reference for saliency map.**
>
> We're so sorry we neglected to add the reference for saliency map.
> We'll add it in the revised version.
>
> **Question3. Ablation studies on the effect of the number of prompting examples.**
>
> We show results on the Gen-T task of RICO in Table 16 (in the attachment).
> Due to limited time and the experiment costs, we run experiments on 1000 test samples.
> The results indicate that the number of prompting examples mainly affects mIoU and FID.
> Specifically, as the number of prompting examples increases, mIoU and FID get improved.
> We'll experiment with the full test set on more tasks, and add the results in the supplementary material.
>
> **Question4. The dataset details.**
>
> We have answered it in Weakness1.
> Please see Weakness1 for the dataset details.
>
> **Question5. Human evaluation.**
>
> We will add a human evaluation in the revised version.
> We would like to point out that most of state-of-the-art work, such as LayoutFormer++ [1], LayoutDM [2] and LDGM [3] do not conduct any human evaluation and we think quantitative and qualitative results are also enough to demonstrate the effectiveness of our method.
>
> [1] LayoutFormer++: Conditional Graphic Layout Generation via Constraint Serialization and Decoding Space Restriction. CVPR2023.
>
> [2] LayoutDM: Discrete Diffusion Model for Controllable Layout Generation. CVPR2023.
>
> [3] Unifying Layout Generation With a Decoupled Diffusion Model. CVPR2023.

---

> > ### Comment · Reviewer_mVCb · 2023-08-10
> >
> > Thanks for providing the rebuttal response along with additional experimental results and more details about the dataset collection.
> >
> > I acknowledge that I have read the response and the additional material in the provided PDF.
> >
> > Overall, I believe this is a sound work and I'm happy to raise my score from 6 to 7.

---

### Official Review · Reviewer_kjZH · 2023-07-06

**Soundness:** 3 good
**Presentation:** 3 good
**Contribution:** 3 good
**Rating:** 6
**Confidence:** 4

**Summary:**

In this paper, the authors introduce LayoutPrompter, a novel approach for conditional graphic layout generation using
large language models with minimal demonstrations.

The authors frame the graphic layout generation task as a sequence generation problem. They represent layouts as HTML
files, leveraging the fact that large language models are pretrained on a corpus that frequently includes HTML. A unique
aspect of this approach is the use of a Dynamic Prompting strategy. In this strategy, the model retrieves exemplars that are
more closely related to each test sample based on a defined constraint similarity metric. This ensures that the in-context
examples used for prompting are highly relevant. Furthermore, a Layout Ranker is appended to the model, which selects
the optimal layout post-inference. Notably, the method does not require extensive training data and demonstrates
adaptability across various layout domains.

To validate the efficacy of LayoutPrompter, the authors carry out experiments across different layout domains. The
performance of LayoutPrompter is benchmarked against various task-specific and task-agnostic baselines. Remarkably,
LayoutPrompter yields competitive results even without domain-specific training or fine-tuning. It is worth noting that
LayoutPrompter is particularly effective on a custom WebUI dataset, which is constructed to investigate a new text-to-layout
generation task. This outperformance is more pronounced when compared to results on other layout domains and
against other baselines.

**Strengths:**

Originality: LayoutPrompter is innovative in using large language models for layout generation without fine-tuning. The
adaptation of dynamically choosing better demos (Dynamic Prompting) and output verification (Layout Ranker) to this
field is a clever utilization of existing techniques.

Quality: The method's solid integration of Sequence Generation, HTML representation, Dynamic Prompting, and Layout
Ranker underpins its robustness. Comprehensive benchmarks add credibility to the research.

Clarity: The paper is well-structured, with clear explanations of methodology and results, making it accessible and easy to
follow.

Significance: LayoutPrompter's ability to generalize without extensive training data is noteworthy, indicating the potential for
diverse applications and research in layout generation.

**Weaknesses:**

Limited Novelty in Techniques: The application of large language models for layout generation is noteworthy, but the utilization of
HTML Representation, Dynamic Prompting, and Layout Ranker lack innovation. It would be interesting to see the exploration or
introduction of new methodologies tailored for this domain.

Inconclusive Ablation Study: The ablation study on the Dynamic Prompting and Layout Ranker does not convincingly demonstrate
the effectiveness of these two techniques.

Insufficient Experimental Analysis: While the paper presents experimental results, it lacks in-depth explanations and analyses.
Elaborating on the implications of these results, and providing a more detailed interpretation, would add substantial value.
In-depth Analysis of Failures: The paper could provide a more detailed analysis of cases where LayoutPrompter fails or
underperforms. Understanding the limitations and areas where the model struggles would offer insights for future research and
improvements.

Writing Clarity and Organization: The paper could improve in terms of clarity and organization. For instance, in discussing the
motivation for bridging the gap between existing methods and human designers, the text lacks coherence and the last two points seem
to overlap. Clearer and more structured presentation of concepts would enhance the paper’s readability.

**Questions:**

Ablation Study Details: Could the authors elaborate on the methodology used for the ablation study, and provide insights into why
Dynamic Prompting and Layout Ranker might not have shown significant effectiveness. Would considering different variations of
these techniques lead to different outcomes?

Detailed Experimental Analysis: Could the authors provide a more detailed analysis of the experimental results? Specifically, why
LayoutPrompter performs better on certain metrics compared to others. Moreover, what are the factors that contribute to its varied
performance across different domains?

Generalization Across Domains: The paper claims that the method can be generalized across domains. Can the authors provide
additional empirical evidence or examples to substantiate this claim? Specifically, how will the model perform in a zero-shot scheme or
given demos in different domains?

Handling of Failures and Underperformance: It would be helpful if the authors could elaborate on the specific cases where
LayoutPrompter fails or underperforms. What are the common characteristics of these cases and are there any strategies that could be
employed to mitigate these issues?

Clarity and Organization of Writing: There are instances where the writing is not very clear or well-organized, as observed in the
explanation of closing the gap between existing methods and human designers. Could the authors clarify the intended message and
possibly consider restructuring this section for better clarity?

**Limitations:**

Yes

---

> ### Author Rebuttal · Authors · 2023-08-10
>
> **Limited novelty in techniques.**
>
> Leveraging large language models for layout generation is by no means an easy task.
> It requires a deep understanding of the layout domain and the according innovations, which can be grounded in our three design choices.
> First, layout representation.
> Due to the inclusion of layout source codes in the training data (e.g., HTML code for Web UI), LLMs have acquired some useful layout knowledge during pre-training.
> Hence, we represent layouts as corresponding HTML codes to better leverage the existing layout knowledge within LLMs.
> Second, ConsSim retrieval criterion in dynamic prompting.
> To help LLMs better understand the constraints, ConsSim is designed to select the examples with the similar constraints in the prompt.
> Third, a layout ranker is proposed to select the layout with the best quality.
> With these three insightful techniques, our LayoutPrompter achieves competitive performance with state-of-the-art approaches.
>
>
> **Ablation study details**
>
> In the ablation study, we primarily analyze the effectiveness of Dynamic Prompting and Layout Ranker.
> To understand the contribution of Dynamic Prompting, we compare LayoutPrompter against its variant (w/o Dynamic Prompting) that performs random sampling for exemplar retrieval and keeps the other components the same as LayoutPrompter.
> Similarly, to understand the effectiveness of Layout Ranker, we compare LayoutPrompter against a variant that randomly picks a layout from an LLM's outputs.
> All the comparisons are conducted on 5 conditional layout generation tasks.
>
> To obtain a more in-depth understanding of Dynamic Prompting, we additionally include 2 studies on the effect of retrieval pool size and the number of exemplars in the prompt.
> Being mindful of the LLM experiment costs and the limited time for rebuttal, we randomly select 1,000 samples from the test set for the experiments.
> The results are in Table 15 and Table 16 (in the attachment), from which we can observe that the pool size and the number of exemplars mainly affect the mIoU and FID metrics.
>
> **Inconclusive Ablation Study**
>
> We kindly argue that Dynamic Prompting and Layout Ranker have shown effectiveness in ablation studies.
> Compared with the variant (w/o Dynamic Prompting), LayoutPrompter achieves significantly better FID and mIoU across the board.
> Though the variant achieves better Alignment and Overlap scores in some tasks, its noticeably poor FID and mIoU scores indicate that it fails to acquire the layout patterns in specific domains (e.g., the generated layout does not look like a real UI layout).
> We also provide the qualitative results of this variant in Figure 19 (in the attachment).
> By comparing Figure 19 and Figure 5, we can conclude that Dynamic Prompting yields better layout quality.
> For Layout Ranker, it consistently yields improvements on the mIoU and Alignment metrics of all tasks.
>
> **Insufficient experimental analysis.**
>
> Due to the black-box nature of deep learning, we cannot precisely tell which factors attribute to the good performance against state-of-the-art methods and the difference in various layout domains.
> Still, we try to discuss two potential factors.
>
> 1) Intrinsic layout complexity of each domain.
> PubLayNet is a document layout dataset, while RICO is a dataset of Android UI.
> Elements in a document layout are typically well-aligned and do not overlap with each other.
> However, overlapping elements in UI are frequently observed (see Figure 15 in the attachment for comparison).
> Such an intrinsic difference may explain why a method often achieves better Alignment and Overlap scores on PubLayNet than RICO.
> Besides, the less aligned and larger overlapping nature of UI layouts may make training-based methods struggle to acquire the layout patterns, which provides a chance for LayoutPrompter to outstand, as it is only exposed to layouts with similar constraints.
>
> 2) Pre-training data distribution.
> It is known that the distribution of the pre-training dataset has a significant impact on large language models' performance on downstream tasks.
> In the publicly available web corpus, the number of Android UI relevant corpus (e.g., Android application source code from Github) may be significantly larger than that of documents.
> As a result, language models are more familiar with Android layouts, which may explain LLMs achieve more state-of-the-art metrics on RICO.
>
> **Generalization across domain.**
>
> For cross-domain, we intend to mean that LayoutPrompter can solve layout generation tasks on different layout domains using the same methodology, rather than performing in a zero-shot scheme or on given demonstrations in different domains.
> We will revise this term.
>
> As suggested, we include the zero-shot results in Figure 17 (in the attachment), from which we can observe that LLMs fail to generate visually pleasing layouts.
>
> **Handling of failures and underperformance.**
>
> One typical failure happens when the number of elements is large.
> This is not a problem specific to our approach and can also be observed in LayoutFormer++.
> We think there are two possible reasons.
> First, it is more difficult to place the elements properly as the number of elements increases.
> Second, the average number of elements in the training set is around 10, which may lead to the issue that existing methods do not perform well when generating layouts whose element number is above average.
>
> **Clarity and organization of writing.**
>
> Thanks for your comments. We will polish the writing.
> Specifically, for the discussion about the gap between existing methods and human designers, we plan to organize it as follows.
> First, the 'high cost' (Line 39) will be explained more clearly as high training cost and reliance on a large amount of data.
> Second, the 'weak cross-domain ability' (line 47) will be revised since it is a bit exaggerated as you pointed out.
> We will put the discussion about layout domains in the point 'relatively task-specific'.

---

### Official Review · Reviewer_HKER · 2023-07-07

**Soundness:** 3 good
**Presentation:** 3 good
**Contribution:** 3 good
**Rating:** 7
**Confidence:** 4

**Summary:**

This paper leverages large language models to solve the layout generation tasks. It shows a great performance when generalizing to other domains or other tasks. Since in-context learning is used, there is no need to train the model on a vast amount of data.

**Strengths:**

1. The prompting method shows a good performance and does not need large amount of training data.
2. The new method can well generalize to new domains and new tasks without an additional model

**Weaknesses:**

1. The paper used text-davinci-003 as LLM, but it is also worthwhile to see how other LLMs perform with the proposed prompt-based method. Can these prompts perform well on other LLMs, including the in-house or open-sourced ones.
2. The in-context learning is not stable with LLMs. The sample prompt may have different results depending on the random seed and temperature parameters. The paper didn't report the variance of the results.
3. In line 258, it is also necessary to mention the parameters for running LLM to repeat the experiments.

**Questions:**

See weaknesses.

**Limitations:**

Yes.

---

> ### Author Rebuttal · Authors · 2023-08-10
>
> **Weakness1. How other LLMs perform with the proposed method.**
>
> Initially, we tried Alpaca 7B.
> However, the model is prone to generate heavily overlapping layouts, and sometimes the output is not even in the right format.
> We then switched to text-davinci-003 and were successful.
> We agree that it is an interesting topic to investigate the performance of other LLMs.
> We will continue to experiment with other powerful LLMs, such as GPT-4 and LLaMa 2, and contribute more techniques and results to the community.
>
> **Weakness2. Stability of LLMs results.**
>
> Since temperature is a trade-off between generation quality and diversity, we used a moderate value of 0.7, which is also the default value of OpenAI API.
> Here, we share our exploration for the impact of random seeds on generating layouts.
> 1. Qualitative results. In Figure 18 (in the attachment), we show the generated layouts (four layouts in each group) under the same constraints but different random seeds.
> Although they look different, they all satisfy the given constraints and are visually pleasing.
>
> 2. Quantitative results.
> To study the effect of random seeds on LLMs, we generate 10 different outputs for each test sample, and then calculate the mean and variance of all metrics (see Table 14 in the attachment).
> We disable Layout Ranker in this experiment to eliminate the impact of post-processing on model performance.
> The small variances show the stability of the proposed method to different random seeds.
>
> Combining these two aspects, we conclude that though LLMs generate different layouts under different random seed, the performance is relatively stable.
>
> **Weakness3. The parameters for running LLMs.**
>
> Thanks for your advice.
> We'll add the parameters for running LLMs in the revised version.
> In fact, for all the parameters, we use default values in OpenAI API.
> For example, $\text{temperature} = 0.7$, $\text{top p} = 1$, $\text{frequency penalty}=0$, $\text{presence penalty}=0$.

---

### Official Review · Reviewer_7S5o · 2023-07-16

**Soundness:** 3 good
**Presentation:** 4 excellent
**Contribution:** 4 excellent
**Rating:** 7
**Confidence:** 5

**Summary:**

The authors propose a new conditional layout generation method using LLMs with few-shot learning. The method uses a pre-trained GPT-3 with two techniques: dynamic prompting and layout ranker. Dynamic prompting is to provide few-shot examples with similar constraints to the input query using ConsSim, and layout ranker is to rank the layout using weighted metrics. The authors show that this method can generate layouts that comply with different types of conditions across domains, and the qualitative and quantitative results show that the method is comparable and sometimes better than existing training-based, domain-specific methods on various metrics.

**Strengths:**

1. The proposed method addressed the problems of existing methods: reducing the training cost using few-shot learning and providing abilities across domains and tasks by properly designed input/output formats and pre-trained LLMs.

2. In addition to generic layout conditions, the proposed method can be applied to two more challenging tasks: content-aware generation and text-to-layout.

3. The two techniques, dynamic prompting and layout ranker, are simple and effective without extra fine-tuning or complex heuristics.

4. The authors demonstrate the possibility of leveraging LLMs for design-related tasks.

**Weaknesses:**

1. It will be great to see more comparisons and discussions on content-aware layout generation. For example, some prior works, such as CanvasVAE [1] and CGL-GAN [2], are missing here.

2. It would be great if the authors could examine if the retrieved examples are similar to the given test set layout. For instance, in RICO, multiple layouts might be extracted from the same app, so they might look very similar and provide a way for the model to cheat.

3. Ablation studies can be improved by adding, for example, zero-shot results, results of different ways to represent the constraints, results of varying screen resolutions, results in groups with different numbers of elements, etc.

4. It would be great if the authors could conduct a user study, ideally on designers, since none of these metrics can perfectly capture human perceptions/judgments.

[1] CanvasVAE: Learning to Generate Vector Graphic Documents, Kota Yamaguchi, ICCV 2021
[2] Composition-aware Graphic Layout GAN for Visual-textual Presentation Designs, Zhou et al., IJCAI 2022



**Questions:**

1. The Gen-T and Gen-TS results in RICO in Fig 5 look very similar, and I wondered if the authors intentionally chose them for comparison. Otherwise, it seems impossible since the conditions are generic and have many possible solutions.

2. In the ablation studies (Table 5), the Gen-T results w/o HTML achieve 0% constraint violations, and the proposed method performs the worst. Are there possible reasons? The other tasks do not have results w/o HTML, and I wondered if the same situation can be observed.

3. It would be interesting to see the results in the refinement task with different perturbation levels.

4. Have the authors tried fine-tuning LLMs on these tasks?

**Limitations:**

Yes, the authors have addressed the limitations.

---

> ### Author Rebuttal · Authors · 2023-08-10
>
> **Weakness1. It will be better to compare with other content-aware layout generation works, such as CanvasVAE and CGL-GAN.**
>
> We clarify that content-aware layout generation refers to generating layouts based on the image content, i.e., avoid putting graphic elements on top of main objects (e.g., faces, animals, and products) and create a pleasing feeling. For CanvasVAE, it can not be applied here directly.
> There are two reasons.
> 1. CanvasVAE is an unconditional VAE model, so canvas/content cannot be input into the model as a condition.
> 2. CanvasVAE generates the entire design with many attributes, such as the element color, opacity and image, which is beyond the scope of layout generation.
>
> Therefore, we don't compare with it in the current paper and leave this to future work.
>
> For CGL-GAN, our baseline DS-GAN has outperformed it on most quantitative and qualitative results. We didn't include it in the paper, and will add it in the revised version.
>
> **Weakness2. The retrieved examples could be similar to the given test set layouts.**
>
> We manually check some test layouts and their corresponding retrieved examples.
> Figure 15 (in the attachment) shows the qualitative results.
> We can observe that the retrieved examples are close to the test layouts, but not exactly the same.
> We think they provide good references to generating different and reasonable layouts.
> In the future, we will try to develop quantitative metrics to make the study along this direction more comprehensive and solid.
>
> **Weakness3. Ablation studies can be improved.**
>
> Thanks for your comments.
> We share the ablation studies we have explored here.
> 1. Zero-shot results.
> The zero-shot qualitative results are shown in Figure 17 (in the attachment).
> It is obvious that LLMs fail to generate reasonable layouts in a zero-shot scheme.
> We'll add zero-shot results in the revised version.
>
> 2. Results of different ways to represent constraints.
> Actually, we have tried different ways to represent constraints but do not find a better way than the one proposed by LayoutFormer++.  Take Gen-T as an example.
> We find that the constraint format "4 images" performs worse than the constraint format "image | image | image | image" used in LayoutFormer++.
> Besides, we find that using a fixed order (e.g., alphabetical order) for element types as LayoutFormer++ can improve performance.
> For example, "image | image | text | text" is better than a random order "image | text | image | text".
>
> 3. Results of varying screen resolutions.
> The four datasets we used in experiments have different screen resolutions.
> On RICO, the screen resolution is discretized to $90px\times160px$.
> PubLayNet has a screen resolution of $120px\times160px$.
> PosterLayout has a screen resolution of $102px\times150px$.
> WebUI has a screen resolution of $120px\times120px$.
>
> 4. Results in groups with different numbers of elements.
> Previous work, such as VTN [1], has found that as the number of elements
> increases, quantitative measures like FID and Overlap become worse.
> We will include a thorough analysis of this in the revised version.
>
> **Weakness4. It would be great to add a user study.**
>
> We will add a user study in the revised version.
> We would like to point out that most of state-of-the-art work, such as LayoutFormer++ [2], LayoutDM [3] and LDGM [4] do not conduct user studies and we think quantitative and qualitative results are also enough to demonstrate the effectiveness of our method.
>
> **Question1. Some results of Gen-T and Gen-TS in RICO are similar.**
>
> It is by coincidence.
> We use Layout Ranker to select the best layout from 10 outputs.
> It chooses the one similar to the result generated by LayoutFormer++ by chance.
> To illustrate that our method can generate diverse layouts, we present other generated layouts (who are not ranked as top 1 by Layout Ranker) in Figure 18 (in the attachment).
> We can observe that they differ from those shown in the paper, and they are also plausible and satisfy the constraints.
>
> **Question2. Possible reasons why the experiment w/o HTML achieves the best constraint violation.**
>
> The one w/o HTML uses the plain sequence to represent the layouts, i.e., $\text{type}_1, x_1, y_1, w_1, h_1, \text{type}_2, x_2 ...$.
> We conjecture that the better constraint violation comes from that the plain sequence is much shorter than HTML code.
> Note that though w/o HTML achieves 0\% constraint violation, it has very pool FID and overlap metrics (see Table 5).
> Differently, the proposed method has negligibly performance drop in terms of constraint violation (from 0\% to 0.64\%) and achieves significantly better quality (e.g., FID and overlap).
> The same result can be observed in Gen-TS, where w/o HTML achieves 0\% constraint violation.
>
> **Question3. It would be better to consider different perturbation levels for refinement task.**
>
> We now follow the perturbation noise in LayoutFormer++.
> We'll try bigger and smaller noises and add the results in the revised version.
>
> **Question4. Whether fine-tuning LLMs is tried.**
>
> We have not tried fine-tuning LLMs on these tasks, since our target is to develop a training-free layout generation method.
> In addition, while there are parameter-efficient fine-tuning techniques for LLMs, they are still much more expensive than a training-free method.
>
> [1] Variational transformer networks for layout generation. CVPR2021.
>
> [2] LayoutFormer++: Conditional Graphic Layout Generation via Constraint Serialization and Decoding Space Restriction. CVPR2023.
>
> [3] LayoutDM: Discrete Diffusion Model for Controllable Layout Generation. CVPR2023.
>
> [4] Unifying Layout Generation With a Decoupled Diffusion Model. CVPR2023.

---

> > ### Comment · Reviewer_7S5o · 2023-08-16
> >
> > Thank the authors for the detailed reply. Most of my questions are addressed. However, I respectfully disagree that current qualitative and quantitative metrics are enough. Since layout design is a professional task, designers can evaluate layouts by considering design rules, aesthetics, styles, usabilities, etc. There are many things that cannot be easily captured by metrics like FID scores. Even in image generation, there have been plenty of existing works discussing the gap between FID/CLIP-Score and human preferences. Therefore, for layout generation to create real world impact, user study is needed. I will keep my current rating.

---

### Author Rebuttal · Authors · 2023-08-10

First, we thank the reviewers for recognizing our simple yet effective work that enables LLMs to generate graphic layouts in a training-free manner. We also appreciate all valuable and constructive comments. We attach our added quantitative and qualitative results here.

---

### Decision · Program_Chairs · 2023-09-21

**Decision:**

Accept (poster)

**Comment:**

The reviewers are generally positive about the paper, and some of them increased their score after the rebuttal. Overall, this is a nice piece of work that leverages LLMs for layout generation, a topic that has gained increasing interest in the field. The work proposes and examines two simple but effective techniques to achieve goals such as cross-domain layout generation. The work has demonstrated the applications in both content-aware generation and text-to-layout tasks. Reviewer 7S5o commended that "The two techniques, dynamic prompting and layout ranker, are simple and effective without extra fine-tuning or complex heuristics." Reviewer kjZH noted that "LayoutPrompter is innovative in using large language models for layout generation without fine-tuning. The adaptation of dynamically choosing better demos (Dynamic Prompting) and output verification (Layout Ranker) to this field is a clever utilization of existing techniques." Reviewer mVCb highlighted the same aspect "The strength of this submission lies in its clear motivation, well-written content, and ease of understanding."

The reviewers do have concerns about the evaluation of the techniques, including ablation, datasets and metrics. For example, Reviewer 7S5o mentioned that "I respectfully disagree that current qualitative and quantitative metrics are enough. Since layout design is a professional task, designers can evaluate layouts by considering design rules, aesthetics, styles, usabilities, etc". Reviewer HKER
 is concerned with the stability of the LLMs results. The authors should incorporate your responses to the reviewers in the revision to further strengthen the paper. In addition, the paper seems to miss a few highly relevant works:

PLay: Parametrically Conditioned Layout Generation using Latent Diffusion, CY Cheng, F Huang, G Li, Y Li, ICML 2023

Auto completion of user interface layout design using transformer-based tree decoders, Y Li, J Amelot, X Zhou, S Bengio, S Si, arXiv preprint arXiv:2001.05308